# Highly efficient dual photoredox/copper catalyzed atom transfer radical polymerization achieved through mechanism-driven photocatalyst design

Woojin Jeon[1], Yonghwan Kwon[1]✉ & Min Sang Kwon [1]✉

Atom transfer radical polymerization (ATRP) with dual photoredox/copper catalysis combines the advantages of photo-ATRP and photoredox-mediated ATRP, utilizing visible light and ensuring broad monomer scope and solvent compatibility while minimizing side reactions. Despite its popularity, challenges include high photocatalyst (PC) loadings (10 to 1000 ppm), requiring additional purification and increasing costs. In this study, we discover a PC that functions at the sub-ppm level for ATRP through mechanism-driven PC design. Through studying polymerization mechanisms, we find that the efficient polymerizations are driven by PCs whose ground state oxidation potential—responsible for PC regeneration—play a more important role than their excited state reducing power, responsible for initiation. This is verified by screening PCs with varying redox potentials and triplet excited state generation capabilities. Based on these findings, we identify a highly efficient PC, 4DCDP-IPN, featuring moderate excited state reducing power and a maximized ground state oxidation potential. Employing this PC at 50 ppb, we synthesize poly(-methyl methacrylate) with high conversion, narrow molecular weight distribution, and high chain-end fidelity. This system exhibits oxygen tolerance and supports large-scale reactions under ambient conditions. Our findings, driven by the systematic PC design, offer meaningful insights for controlled radical polymerizations and metallaphotoredox-mediated syntheses beyond ATRP.

Atom transfer radical polymerization (ATRP) provides a direct pathway to a multitude of precisely defined (co)polymers with predetermined molecular weights (MWs), narrow MW distributions, and a significant degree of chain end functionality[1–6]. Notably, as a radical process, this method exhibits tolerance towards various functional groups and can be performed under relatively mild conditions. Consequently, it has found efficient applications to produce polymers with precisely controlled functionalities, topologies, and compositions[7,8].

ATRP regulates the reaction by an equilibrium between active and dormant species mediated by activator and deactivator forms of copper catalyst, based on the persistent radical effect[9,10]. In the early stages of ATRP development, Cu(I)Br/L (L = ligand) was predominantly used as a starting catalyst[11,12]. However, as the method evolved, reverse ATRP emerged, utilizing the highly stable and cost-effective Cu(II)Br$_2$/L as the catalyst precursor[13–15]. Subsequently, through the regeneration of accumulated Cu(II)Br$_2$/L—caused by irreversible bimolecular radical

[1]Department of Materials Science and Engineering and Research Institute of Advanced Materials, Seoul National University, Seoul, Republic of Korea.
✉e-mail: yhkwon1995@snu.ac.kr; minsang@snu.ac.kr

terminations and acting as the deactivator–into Cu(I)Br/L through external stimulus, the required catalyst amount was dramatically reduced to the parts per million (ppm) level[16–18]. Particularly noteworthy is the pioneering research conducted by Yagci's group[19–25], Haddleton's group[26–28], and Matyjaszewski's group[29–31]. Their works in utilizing light for ATRP (photo-ATRP) not only significantly reduce the required amount of copper catalyst but also enable spatiotemporal control, exhibit a certain degree of oxygen tolerance, and operate at room temperature. This has prompted a surge of recent research in this area[24,25]. However, within this method, the use of light is confined to high-energy UV light, which potentially leads to insufficient controllability and undesired side reactions.

Photoredox-mediated ATRP, subsequently developed by the group of Hawker[32,33], Miyake[34,35], and Matyjaszewski[36,37], establishes a method to active dormant species through the outer-sphere electron transfer (ET) with an organic or organometallic photocatalyst (PC). By carefully designing the PCs[38–40], the range of light capable of initiating the reaction can be expanded to include visible light, and the PC loadings can also be reduced to the level of ppm[38]. For example, the groups led by Zhu and Miyake reduced PC loadings to approximately 10 ppm for the synthesis of poly(methyl methacrylate) (PMMA)[41,42]. Later, the groups of Kwon and Liao further reduced PC loadings to sub-ppm levels[38,43]. However, the range of applicable monomers, solvents,

and MWs remains relatively limited. Moreover, although there are some recent reports on oxygen tolerance[42,44], most studies report no such tolerance. Notably, the deactivation process essentially involves three molecules[36], which is recognized as a slight disadvantage compared to the conventional ATRP method. Furthermore, external factors such as solvents can influence this process[45,46]. In addition, potential issues concerning the PC's stability may arise due to undesired side reactions between the PC and propagating radicals[47].

ATRP with dual photoredox/copper catalysis has recently been proposed by the groups of Yagci[48–50], Strehmel[50–52], and Matyjaszewski[53–60]. This approach combines the advantages of both photo-ATRP and photoredox-mediated ATRP. It employs the organic or organometallic PC capable of light absorption in the visible range as a photosensitizer, while $Cu(II)Br_2/L$ functions as the deactivator to regulate ATRP equilibrium. Currently, roughly three possible mechanisms for this method have been proposed. In the first scenario, the PC in an photoexcited state reduces $Cu(II)Br_2/L$ to $Cu(I)Br/L$, initiating the reaction. The concentration of radicals is then governed by the ATRP equilibrium between $Cu(I)Br/L$ and $Cu(II)Br_2/L$ (Fig. 1a, Mechanism I). As the ligand undergoes oxidation, it facilitates the regeneration of PC[53–58]. $Cu(I)Br/L$ and propagating radical species, along with bromide, can also promote PC regeneration through termolecular processes. However, considering that PC regeneration via

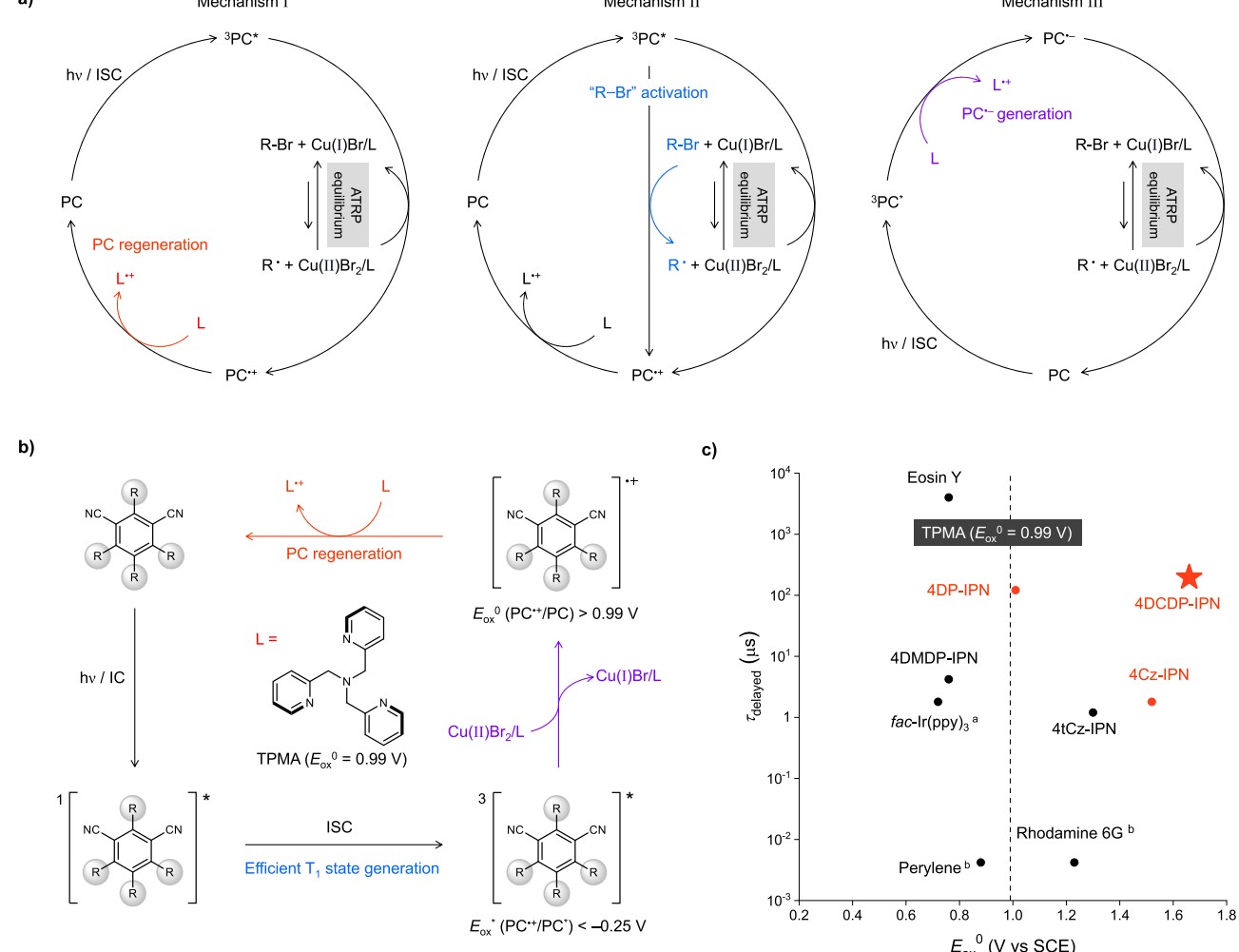

**Fig. 1 | Schematic summary of this work. a** Previously reported mechanisms of atom transfer radical polymerization (ATRP) with dual photoredox/copper catalysis; mechanism I (left), mechanism II (center), mechanism III (right). Here, PC, L, and ISC denote photocatalyst, ligand, and intersystem crossing, respectively. **b** Proposed mechanism of this work utilizing a cyanoarene-based PC in a triplet excited state. Here, the IC and $T_1$ state denote internal conversion and the lowest triplet excited state, respectively. **c** Ground state oxidation potential ($E_{ox}^0$) and delayed fluorescence (DF) lifetime ($\tau_{delayed}$) of PCs are illustrated. Otherwise, the PL lifetime of PCs is indicative of [a]phosphorescence or [b]fluorescence lifetime, respectively.

ligand oxidation involves a bimolecular reaction, and noting the ligand's significantly higher concentrations—due to its excess use—relative to Cu(I)Br/L and propagating radical species, it is broadly recognized that ligand oxidation serves as the dominant mechanism for PC regeneration; for a more detailed discussion of the mechanism, please refer to the supplementary information (Supplementary Fig. 14). In the second mechanism, an additional process involves the photoexcited PC directly participating in the activation of dormant species (Fig. 1a, Mechanism II)[61,62]. In the other mechanism, PC in the photoexcited state is initially reduced via ET process with the ligand, forming the PC radical anion (PC$^{\bullet-}$) (Fig. 1a, Mechanism III)[59,60]. Subsequently, the PC$^{\bullet-}$ facilitates the reduction of Cu(II)Br$_2$/L to Cu(I)Br/L, simultaneously regenerating the ground state of PC. The reaction mechanism depends on the types of PCs and ligands used. Specifically, when methyl methacrylate (MMA) serves as a monomer, tris(2-pyridylmethyl)amine (TPMA) is commonly chosen as the ligand[14]. Considering the ground state reduction potential ($E_{red}^0$) of Cu(II)Br$_2$/L ($E_{red}^0 = -0.25$ V with L = TPMA) and ground state oxidation potential ($E_{ox}^0$) of TPMA ($E_{ox}^0 = 0.99$ V), it is important to note that most reactions typically proceed via either Mechanism I or II; we thus here focus on systems operating under these mechanisms (see below).

Indeed, this method enables well-controlled polymerization across a wider range of monomer groups due to i) its incorporation of the bimolecular deactivation process via Cu(II)Br$_2$/L, similar to conventional ATRP, and ii) utilization of visible light as an excitation source. Notably, this method has been reported to exhibit significantly higher oxygen tolerance compared to the conventional ATRP. The mechanism for remarkable oxygen tolerance is believed to rely on the oxygen scavenging ability of Cu(I)Br/L and/or the lowest triplet excited (T$_1$) state of PC[55,59]. This leads to substantial improvements in the practical utility of ATRP[59,60,63–65].

Despite recent advancements in the ATRP with dual photoredox/copper catalysis, the PC loadings still remain relatively high, typically ranging from 10 to 1000 ppm[50], necessitating additional purification, hindering scalability, and potentially increasing overall production costs. These issues likely arise from the laborious process of PC discovery, which predominantly relies on trial-and-error methods, rather than being guided by a systematic design. This involves modifying and adapting PCs that have shown efficient in analogous photo-mediated controlled radical polymerization (CRP) methods such as organocatalyzed ATRP (O–ATRP)[61] and/or photoinduced electron/energy transfer reversible addition-fragmentation chain-transfer (PET–RAFT) polymerization[55]. A central issue is the inadequate consideration of the mechanistic aspects of dual photoredox/copper catalysis in the PC design. In the mechanisms, the reduction of Cu(II)Br$_2$/L is the most crucial aspect, as it governs the initiation of the reaction and ATRP equilibrium. Given the low $E_{red}^0$ of Cu(II)Br$_2$/L, a high reducing power of the PC in the excited state is unnecessary; indeed, most PCs already meet this condition. Conversely, it becomes crucial to maximize $E_{ox}^0$ of PC to facilitate its efficient regeneration through ET from the ligand. However, the development of PCs by numerous research groups continues to focus on activating dormant species and/or Cu(II)Br$_2$/L by the PC, mostly neglecting the significance of PC regeneration (Supplementary Table 1 and Supplementary Fig. 4).

In this study, we have developed a PC that functions at the sub-ppm level for ATRP with dual photoredox/copper catalysis, with a primary focus on promoting the PC regeneration process (Fig. 1b, c). To minimize PC loading, the PC was designed employing a donor−acceptor (D−A) cyanoarene molecular scaffold, which facilitates the straightforward modulation of the redox potentials through modifications of D and/or A, while also efficiently generating the long-lived T$_1$ state. Given the ligand's relatively high $E_{ox}^0$ value, the energy level of the highest occupied molecular orbitals (HOMO) of PC was intentionally lowered to maximize its $E_{ox}^0$, thus facilitating fast regeneration through the oxidation of the ligand. Because of the presence of the electron-withdrawing group (i.e., −CN group) substituted at the donor, the lowest unoccupied molecular orbital (LUMO) is positioned at a relatively low energy level. As a result, the reducing power of this PC in the excited state is relatively modest when compared to other previously used PCs, rendering it insufficient to directly activate the ATRP initiator (i.e., ethyl α-bromophenylacetate, EBPA; see Mechanism II); however, it efficiently reduces Cu(II)Br$_2$/L to Cu(I)Br/L. To verify the validity of our PC design principle, we prepared nine different PCs with varying the ground- and excited state oxidation potentials ($E_{ox}^0$ and $E_{ox}^*$, respectively), T$_1$ state generation, and T$_1$ state lifetime, for a comparison. The discovered PC, 4DCDP-IPN, demonstrated high catalytic activity with the optimized combination of Cu(II)Br$_2$ and ligand. During the polymerization of MMA, this PC exhibited decent performance even at 50 ppb (parts per billion), a narrow MW distribution, and an oxygen tolerance. It also showed good control over various methacrylate-based monomers and enabled the successful synthesis of block copolymers containing styrene monomer. Moreover, this low PC loading allows for comparably deep light penetration to the center of the reaction batch, leading to a more accomplished reaction[66]. Combining this with the oxygen tolerance, at the same PC loading of 50 ppb, we conducted large-scale polymer synthesis without a need for oxygen removal, even with low loading of Cu(II)Br$_2$/L (10 ppm). We believe these results represent more than just a development of the efficient PC. Serving as a meaningful example of mechanism-driven PC design, it establishes a strong foundation for addressing various challenges related to dual photoredox/copper catalysis in the future.

## Results

### Preparation and characterization of PCs

4DCDP-IPN was synthesized by modifying 4DP-IPN with diphenylamine serving as the donor moiety (Fig. 2). 4DP-IPN, an organic PC, is known for its efficacy in various light-driven organic reactions[67–75], and polymerizations[38,76–81], due to its ability to efficiently generate long-lived T$_1$ state ($\tau_{delayed} = 121.5$ μs, $\Phi_{ISC} = 0.84$) and exhibit appropriate redox potentials. To minimize the HOMO of PC, 4,4′-dicyanodiphenylamine (i.e., DCDP) was chosen as a donor moiety. This PC features a strong electron-withdrawing −CN group substituted at the para position of diphenylamine. For comparison, a set of four different D−A cyanoarenes were also prepared (i.e., 4DMDP-IPN, 4DP-IPN, 4tCz-IPN, and 4Cz-IPN; Fig. 2); a comprehensive description of their synthetic approaches is provided in section 1.6 of the Supplementary Information. These PCs vary in terms of the oxidation potentials (i.e., $E_{ox}^0$ and $E_{ox}^*$), T$_1$ state generation, and T$_1$ state lifetime. Additionally, we included conventional PCs like Eosin Y[55–57], Rhodamine 6 G, *fac*-Ir(ppy)$_3$[61], and Perylene[61] as reference PCs; these compounds have been previously used as PCs in ATRP with dual photoredox/copper catalysis[55–57,61].

We initially evaluated $E_{ox}^0$, associated with PC regeneration, and $E_{ox}^*$ of PC, which is correlated to the activation of Cu(II)Br$_2$/L or dormant species, using electrochemical/photophysical measurements; the electrochemical and photophysical properties are summarized and schematically presented in Fig. 2. In general, $E_{ox}^0$ is determined by the HOMO of the PC, while $E_{ox}^*$ is determined by the LUMO. In the case of D−A cyanoarene, the HOMO is located in the donor and the LUMO in the acceptor moiety, consequently determining $E_{ox}^0$ generally by the donor and $E_{ox}^*$ by the acceptor. When methoxy (−OCH$_3$), a strong electron-donating group, is substituted on the diphenylamine, specifically in 4DMDP-IPN, the HOMO becomes much higher, leading to a decrease in $E_{ox}^0$ compared to 4DP-IPN. Simultaneously, the LUMO increases significantly due to the inductive effect, which originates from enhanced the electron-donating power of the donor part, resulting in a more negative $E_{ox}^*$. Conversely, in 4DCDP-IPN, both HOMO and LUMO decrease due to the effect of the −CN group, leading to a significant increase in both $E_{ox}^0$ and $E_{ox}^*$. 4Cz-IPN, featuring carbazole substituted for the diphenylamine moiety, exhibits a less

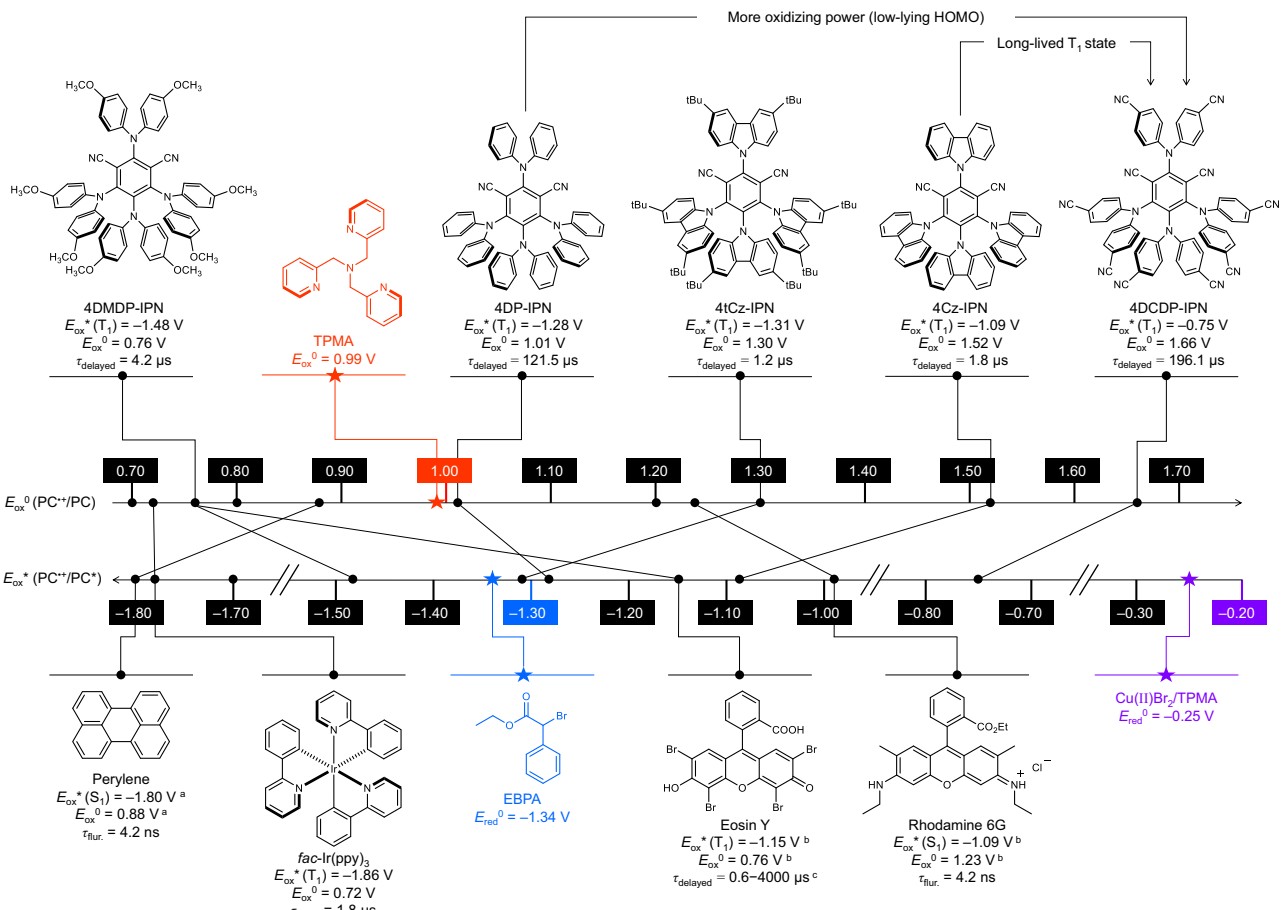

**Fig. 2 | Comparison of the properties of PCs.** Chemical structures and properties of PCs and reagents in ATRP. $E_{ox}^0$ of PCs values were measured against saturated calomel electrode (SCE) in $CH_3CN$ using cyclic voltammetry (CV). The excited state oxidation potentials ($E_{ox}^*$) of PCs were estimated from $E_{ox}^* = E_{ox}^0 - E_{00}$; $E_{00}(T_1)$ was evaluated from the onset of gated PL emission in $CH_3CN$ at 65 K. For $fac$-Ir(ppy)$_3$, $E_{00}(T_1)$ was evaluated from the onset of PL emission in $CH_3CN$ at r.t. [a,b]Redox potentials of [a]Perylene[61], [b]Eosin Y[82], and Rhodamine 6G[82] were referred to literature. [c]$\tau_{delayed}$ of Eosin Y was referred to literature[84,85].

pronounced electron-donating effect owing to increased conjugation within the carbazole, resulting in lower HOMO and LUMO levels. Therefore, compared to 4DP-IPN, it shows higher $E_{ox}^0$ and $E_{ox}^*$. In 4tCz-IPN, t-butyl—a weak electron-donating group—is substituted to the carbazole, leading to a slight increase in HOMO and LUMO levels, and a slight decrease in $E_{ox}^0$ and $E_{ox}^*$ compared to 4Cz-IPN. This trend in the redox potentials aligns well with quantum chemical (QC) calculations (Supplementary Figs. 11 and 12). $E_{ox}^*$ and $E_{ox}^0$ of Eosin Y[82], Rhodamine 6 G[82], $fac$-Ir(ppy)$_3$, and Perylene[61]—which were used as the reference PCs—were either measured or taken from the literature. Compared to cyanoarene-based PCs, $E_{ox}^*$ is generally sufficiently low and $E_{ox}^0$ is less positive.

Next, we measured the photophysical properties of five cyanoarene PCs using UV/vis absorption and photoluminescence (PL) emission spectra in $CH_3CN$ (Supplementary Table 2 and Supplementary Figs. 5 and 6). These cyanoarene PCs exhibited a broad visible-light absorption with a pronounced charge-transfer band. Subsequently, the decays of their PL were monitored with time-correlated single photon counting (TCSPC) techniques. While 4DMDP-IPN ($\tau_{delayed} = 4.2\,\mu s$), 4tCz-IPN ($\tau_{delayed} = 1.2\,\mu s$), and 4Cz-IPN ($\tau_{delayed} = 1.8\,\mu s$) showed relatively short delayed fluorescence (DF) lifetimes, 4DP-IPN ($\tau_{delayed} = 121.5\,\mu s$) and 4DCDP-IPN ($\tau_{delayed} = 196.1\,\mu s$) exhibited significantly longer DF lifetimes exceeding 100 $\mu s$. This indicates that such PCs possess a notably higher $T_1$ state concentration under a photostationary state compared to other PCs, suggesting a correspondingly faster rate of ET process, and

consequently, a reduction in PC loadings[39,68,76,83]. The reference PCs each exhibit an PL emission lifetime that is significantly shorter than those of the cyanoarene PCs ($\tau_{delayed}$ (Eosin Y) = 0.6–4000 $\mu s$)[84,85], $\tau_{flur.}$ (Rhodamine 6G) = 4.2 ns, $\tau_{phos.}$ ($fac$-Ir(ppy)$_3$) = 1.8 $\mu s$, and $\tau_{flur.}$ (Perylene) = 4.2 ns).

**Screening PCs in ATRP with photoredox/copper dual catalysis**

We then performed the polymerization of MMA using nine types of prepared PCs (Table 1). Our experimental procedure was based on the previously established conditions (Supplementary Table 1), optimization experiments (Supplementary Tables 6–8), and reproducibility tests (Supplementary Table 11). The procedure included the use of MMA as the monomer, EBPA as the ATRP initiator at a ratio of 200 relative to the monomer, Cu(II)Br$_2$ added at 10 ppm, TPMA employed as the ligand at 45 ppm, and $N,N$-dimethylformamide (DMF) as a solvent. The reactions were carried out under a 455 nm irradiation with an intensity of approximately 50 mW cm$^{-2}$. A control experiment without a PC was first conducted, resulting in a monomer conversion of approximately 20%, which aligns well with the previous works (Table 1, entry 1)[61,86]. A recent study also reported the significant conversion even in the absence of a PC, attributing this to impurities in the commercially available TPMA[86]. We concluded that the presence of these impurities would not significantly affect a trend in the reactivities of the screened PCs; therefore, we did not proceed with any additional purification processes for TPMA. Table 1 presents the results of MMA polymerization using the prepared PCs at various loadings.

**Table 1 | Results of the polymerizations with various PCs**

PC, MMA
Cu(II)Br$_2$ (10 ppm) / TPMA (45 ppm)
DMF, r.t., 24 h, N$_2$, 455 nm (50 mW cm$^{-2}$)

| Entry | PC | PC loading (ppm) | Cu(II)Br$_2$ (ppm) | α (%) | M$_{n,theo}$ (g mol$^{-1}$)[c] | M$_{n,exp}$ (g mol$^{-1}$)[d] | I*[e] | Đ[f] |
|---|---|---|---|---|---|---|---|---|
| 1 | – | – | 10 | 20 | 4300 | 5300 | 0.81 | 1.36 |
| 2[a] | Eosin Y | 50 | 10 | 95 | 19,200 | 22,000 | 0.87 | 1.24 |
| 3[a] | | 10 | 10 | 25 | 5200 | 7200 | 0.73 | 1.28 |
| 4[a] | | 1 | 10 | No polymerization | | | | |
| 5[a] | Rhodamine 6G | 10 | 10 | 91 | 18,500 | 18,700 | 0.99 | 1.23 |
| 6[a] | | 1 | 10 | 44 | 9100 | 10,700 | 0.85 | 1.23 |
| 7[a] | | 0.1 | 10 | No polymerization | | | | |
| 8 | fac-Ir(ppy)$_3$ | 1 | 10 | 66 | 13,400 | 15,900 | 0.84 | 1.22 |
| 9 | | 0.1 | 10 | 37 | 7,600 | 9,600 | 0.79 | 1.27 |
| 10 | Perylene | 1 | 10 | 57 | 11,700 | 14,100 | 0.83 | 1.22 |
| 11 | | 0.1 | 10 | 24 | 5000 | 6900 | 0.72 | 1.29 |
| 12 | 4DMDP-IPN | 1 | 10 | 61 | 12,500 | 14,000 | 0.89 | 1.22 |
| 13 | | 0.1 | 10 | 20 | 4300 | 5000 | 0.86 | 1.40 |
| 14 | 4DP-IPN | 1 | 10 | 97 | 19,700 | 20,800 | 0.95 | 1.31 |
| 15 | | 0.1 | 10 | 71 | 14,500 | 15,400 | 0.94 | 1.20 |
| 16 | | 0.05 | 10 | 46 | 9500 | 11,700 | 0.81 | 1.24 |
| 17 | 4tCz-IPN | 1 | 10 | 75 | 15,200 | 15,400 | 0.99 | 1.34 |
| 18 | | 0.1 | 10 | 53 | 10,800 | 10,500 | 1.03 | 1.27 |
| 19 | | 0.05 | 10 | 14 | 3000 | 4000 | 0.76 | 1.27 |
| 20 | 4Cz-IPN | 1 | 10 | 96 | 19,500 | 20,800 | 0.94 | 1.28 |
| 21 | | 0.1 | 10 | 60 | 12,200 | 12,200 | 1.00 | 1.22 |
| 22 | | 0.05 | 10 | 58 | 11,800 | 11,100 | 1.06 | 1.32 |
| 23 | 4DCDP-IPN | 1 | 10 | 89 | 18,100 | 20,300 | 0.89 | 1.35 |
| 24 | | 0.1 | 10 | 85 | 17,200 | 17,900 | 0.96 | 1.21 |
| 25 | | 0.05 | 10 | 71 | 14,400 | 16,700 | 0.86 | 1.20 |
| 26[b] | | 0.05 | 10 | 69 | 14,100 | 19,200 | 0.74 | 1.22 |

Polymerizations were carried out under the irradiation of 455 nm (50 mW cm$^{-2}$) for 24 h at r.t. following the general procedure with condition, [MMA]$_0$:[EBPA]$_0$:[PC]$_0$:[Cu(II) Br$_2$]$_0$:[TPMA]$_0$ = [200]:[1]:[x]:[0.002]:[0.009], MMA/DMF = 1/1 (v/v) where x = 1 x 10$^{-2}$, 2 x 10$^{-3}$, 2 x 10$^{-4}$, 2 x 10$^{-5}$ or 1 x 10$^{-5}$ (for 50 ppm, 10 ppm, 1 ppm, 0.1 ppm and 50 ppb relative to the monomer, respectively). Monomer conversion (α) was determined by integrals of $^1$H NMR peaks. [a,b]Reaction was performed under [a]the irradiation of 515 nm (50 mW cm$^{-2}$) and [b]undegassed conditions with 2 mL scale using into a 4 mL vial. [c]M$_{n,theo}$ was determined by following relationship, M$_{n,theo}$ = [MMA]$_0$/[EBPA]$_0$ × conversion × M$_{n,MMA}$ + M$_{n,EBPA}$. [d]M$_{n,exp}$ was measured by gel permeation chromatography (GPC) equipped with refractive index detector calibrated with PMMA standards. [e]Initiator efficiency (I*) was determined by I* = M$_{n,theo}$/M$_{n,exp}$. [f]Dispersity (Đ) was determined by Đ = M$_{w,exp}$/M$_{n,exp}$.

Interestingly, the results showed a notable correlation with $E_{ox}^0$. The five PCs (i.e., 4DP-IPN, 4tCz-IPN, 4Cz-IPN, 4DCDP-IPN, and Rhodamine 6G) with $E_{ox}^0$ values greater than that of TPMA exhibited better performances, particularly at loadings of 1 ppm or lower. Moreover, the difference became more noticeable as the PC concentration decreased. In contrast, the four PCs with lower $E_{ox}^0$ values than TPMA—specifically, Eosin Y, fac-Ir(ppy)$_3$, Perylene, and 4DMDP-IPN—demonstrated somewhat inferior results, despite their $E_{ox}^*$ values being negative enough to directly activate EBPA. These results strongly suggest that in the ATRP with dual photoredox/copper catalysis, the direct activation of EBPA may not be essential, while the regeneration of PCs could be crucial, as anticipated. The results from the reactions involving 4Cz-IPN and fac-Ir(ppy)$_3$ further support this hypothesis. fac-Ir(ppy)$_3$ ($\Phi_{ISC}$ = 1.00)[87] is more efficient than 4Cz-IPN ($\Phi_{ISC}$ = 0.041)[88] at generating the T$_1$ state, and its excited state also possesses significantly greater reducing power ($E_{ox}^*$ = −1.86 V for fac-Ir(ppy)$_3$). However, despite more efficient initiation through faster ET with EBPA and/or Cu(II)Br$_2$/L, it provides inferior results at lower PC loadings (Supplementary Tables 9 and 10). This inefficiency is likely due to the insufficient $E_{ox}^0$ of fac-Ir(ppy)$_3$ ($E_{ox}^0$ = 0.72 V), which leads to inefficient PC regeneration at low loadings. Consequently, PC depletion would

occur, hindering the reaction. Similarly, Rhodamine 6G appears to be a more efficient PC at lower loadings compared to Eosin Y (Supplementary Table 9).

In particular, the polymerizations involving three PCs, namely 4DP-IPN, 4Cz-IPN, and 4DCDP-IPN, proceeded efficiently even at very low loadings. Notably, 4DCDP-IPN demonstrated an excellent monomer conversion as 71% and narrow MW distribution (Đ = 1.20) at a remarkably low loading of 50 ppb. However, 4tCz-IPN, despite its significantly large $E_{ox}^0$, showed poor performance at low loadings compared to 4DP-IPN, which has a relatively smaller $E_{ox}^0$. This is presumed to be originated from a low PL quantum yield ($\Phi_F$ = 0.05) and short DF lifetime ($\tau_{delayed}$ = 1.2 μs) of 4tCz-IPN, as indicated in Supplementary Table 2. 4tCz-IPN exhibits the significantly lower PL quantum yield compared to 4Cz-IPN ($\Phi_F$ = 0.18), 4DP-IPN ($\Phi_F$ = 0.61), and 4DCDP-IPN ($\Phi_F$ = 0.61), suggesting that a considerable population of excited states are lost through non-radiative decay from S$_1$ to S$_0$, consequently inhibiting the overall ET rate. Meanwhile, among the various PCs tested, Eosin Y and Rhodamine 6G exhibited particularly poor results at low loadings, likely due to the PC depletion (Supplementary Fig. 21). In fact, during the polymerization process, Eosin Y experienced the notably rapid PC degradation[89–92], necessitating a significantly higher PC loadings than

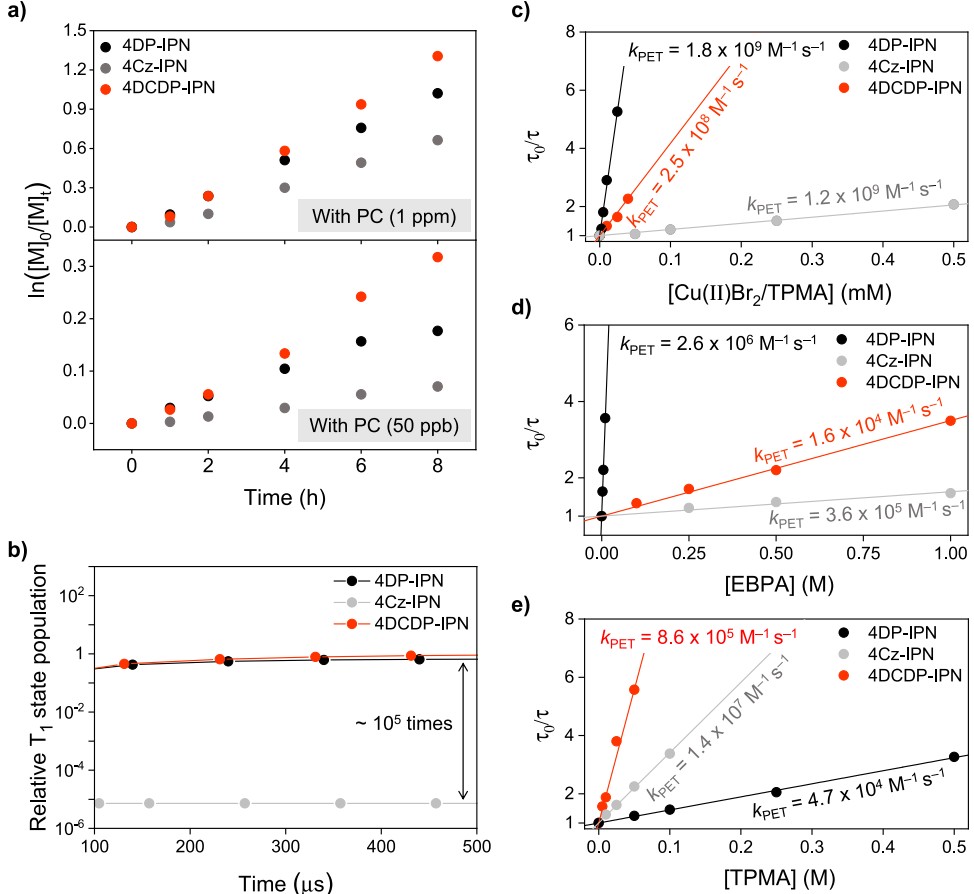

**Fig. 3 | Mechanistic study in ATRP with photoredox/copper dual catalysis.**
**a** Kinetic plots over time for ATRP for the synthesis of PMMA with cyanoarene-based PCs at 1 ppm (top) and 50 ppb (bottom). Polymerizations were conducted following the general procedures. **b** Results of kinetics simulation of the relative $T_1$ state populations for three cyanoarene-based PCs: 4DP-IPN (black line), 4Cz-IPN (gray line), and 4DCDP-IPN (red line) in $CH_3CN$ ($4.65 \times 10^{-6}$ M for 1 ppm in the actual polymerizations) over time under the continuous 455 nm irradiation (50 mW cm$^{-2}$)

(see section 2.4 of the Supplementary Information for more detail). **c**–**e** Stern-Volmer plots for DF decay quenching of degassed solution of PCs in DMF ($1.0 \times 10^{-5}$ M) at r.t. along with addition of (**c**) Cu(II)Br$_2$/TPMA (mixture with 1:1 molar ratio of Cu(II)Br$_2$ to TPMA), (**d**) EBPA, and (**e**) TPMA. DF decay was monitored using time-correlated single photon counting (TCSPC) techniques at $\lambda_{ex} = 377$ nm and $\lambda_{det} = 520$ nm (for 4DP-IPN), 548 nm (for 4Cz-IPN), and 530 nm (for 4DCDP-IPN), respectively.

other PCs for the polymerization to proceed[55–57]. Conversely, in the case of the cyanoarene PCs, it was evident that they persisted even at the end of the reaction. This was confirmed by monitoring the changes in PL emission of 4DP-IPN, 4Cz-IPN, and 4DCDP-IPN (Supplementary Figs. 22–24). To ensure the visibility of their PL emissions, we fixed the PC loading at 1 ppm. Although the PL intensities of all three PCs diminished compared to their initial stages, their PL emissions were still observable to the naked eye. This suggests that substantial amounts of PC remained and continued to drive the reaction. This is further supported by the observed increase in the monomer conversion over time, as illustrated in Figs. 3a and 4a. It have been previously reported that the degradation of cyanoarene PCs can occur via (i) a substitution of the −CN group with alkyl radicals[68,93], and (ii) a self-degradation upon the photoexcitation, such as a intramolecular cyclization[94] or chemical bond-cleavages[95]. In our reaction, given the very low radical concentration controlled by the ATRP equilibrium, their reactivities with the PC species (i.e., PC·− or PC·+) are likely suppressed. However, understanding of the overall PC degradation behaviors are still unclear; in-depth study by our group is currently underway.

## Mechanistic aspect in ATRP with photoredox/copper dual catalysis

To enhance our understanding of how PCs' properties influence polymerizations, we conducted further studies on three PCs− 4DP-

IPN, 4Cz-IPN, and 4DCDP-IPN—which exhibited exceptional performances. Initially, to assess the reactivities of these PCs at low loadings in detail, polymerization kinetics experiments were carried out at 1 ppm and 50 ppb. As depicted in Fig. 3a, the polymerization reaction catalyzed by 4DCDP-IPN demonstrated a fastest kinetics in the monomer conversion, exceeding those mediated by 4DP-IPN and 4Cz-IPN. These kinetics suggest a relatively higher concentration of radical species in the reaction utilizing 4DCDP-IPN compared to those with the other two PCs, indicating that 4DCDP-IPN is a more efficient PC.

We then investigated the polymerization mechanism governed by each PC. This exploration involved examining the rate of electron transfer ($\nu_{ET}$), which is described by the following equation:

$$\nu_{ET} = k_{ET}[PC^*][Q] \tag{1}$$

Here, $k_{ET}$ represents the rate constant of ET, $[PC^*]$ denotes the concentration of PC in the excited state, and $[Q]$ refers to the concentration of substrate of interest. Consequently, understanding the ET rate between the PCs and the substrate requires information on the concentration of the PCs in its excited state during the reaction. This can be determined by a kinetics simulation of the relative concentrations of the $T_1$ state of 4DP-IPN, 4Cz-IPN, and 4DCDP-IPN under the photostationary state (Fig. 3b, See section 2.4 of the Supplementary

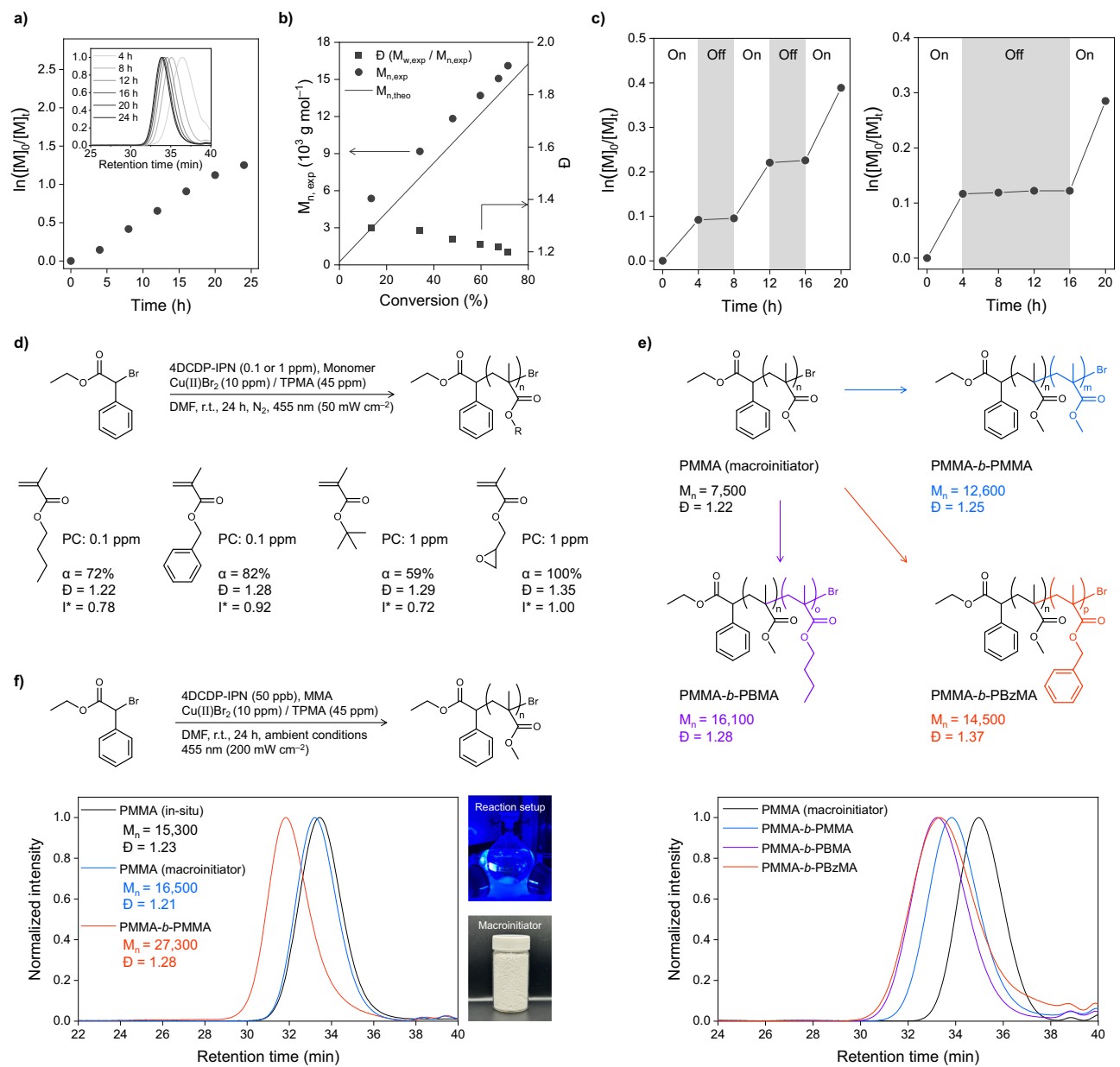

**Fig. 4 | Control and scalability in ATRP with photoredox/copper dual catalysis.**
**a** Kinetic plots over time for the polymerization of MMA. Polymerization was conducted following the general procedures with a target degree of polymerization (DP) = 200. The GPC traces of each in-situ sample are also given (inset). **b** Plot of $M_n$ and Đ variations over the monomer conversion. **c** Temporal control of the synthesis of PMMA by switching the irradiation on and off. **d** Validation of methacrylate-based monomers scope. Polymerizations were conducted under the optimized reaction conditions with 4DCDP-IPN at 1 ppm or 0.1 ppm. **e** Results of block copolymerization with 4DCDP-IPN at 50 ppb. After the preparation of PMMA macroinitiator (black) with 4DCDP-IPN at 50 ppb with a target DP = 100, block copolymerizations were conducted with MMA, BMA, and BzMA with 4DCDP-IPN at 50 ppb with a target DP = 200. GPC traces of the macroinitiator (black) and

synthesized block copolymers PMMA-*b*-PMMA (blue), PMMA-*b*-PBMA (purple), and PMMA-*b*-PBzMA (red) are also given (bottom), respectively. **f** Large-scale polymerization results under the ambient conditions with 4DCDP-IPN at 50 ppb. The large-scale polymerization was carried out under the irradiation of 455 nm (200 mW cm⁻²) for 24 h at r.t. without any degassing process. Its total volume was 200 mL (MMA/DMF = 1/1 (v/v) with a target DP = 200. A subsequent chain-extension experiment was conducted with 4DCDP-IPN at 50 ppb with a target DP = 400. GPC traces of in-situ samples in the large-scale polymerization (black), the isolated macroinitiator (blue), and the synthesized PMMA-*b*-PMMA (red) are also given. Images of large-scale reaction setup and isolated macro-initiators are also given (right).

Information for the details). By utilizing the photophysical properties of these PCs obtained from the experiments and/or literature (for 4Cz-IPN)[88], we simulated the population of each PC in $T_1$ state. As illustrated in Fig. 3b, under the same light wavelength (455 nm) and intensities (50 mW cm⁻²), 4DP-IPN and 4DCDP-IPN are capable of generating approximately $10^5$ times more concentrated population of $T_1$ state than that of 4Cz-IPN. This marked difference in the population of their excited states might explain why 4DP-IPN and 4DCDP-IPN, compared

to 4Cz-IPN, exhibit superior catalytic performances at low PC loadings, despite having worse or similar $E_{ox}^0$, respectively.

We next measured the rate constants of ET, specifically by monitoring the changes in DF decay of the three PCs upon the addition of 1:1 mixture of Cu(II)Br₂ and TPMA, and EBPA, using the TCSPC technique. Figure 3c, d show the Stern-Volmer plots for the three PCs with Cu(II)Br₂/TPMA or EBPA. For 4Cz-IPN and 4DCDP-IPN, the rate constants with Cu(II)Br₂/TPMA ($k_{ET}$ = 1.2 x 10⁹ M⁻¹ s⁻¹ and

$k_{ET}$ = 2.5 x $10^8$ $M^{-1}$ $s^{-1}$, respectively) were approximately $10^4$ times higher than with EBPA ($k_{ET}$ = 3.6 x $10^5$ $M^{-1}$ $s^{-1}$ and $k_{ET}$ = 1.6 x $10^4$ $M^{-1}$ $s^{-1}$, respectively). In contrast, for 4DP-IPN, the increase was about $10^2$ to $10^3$ times. Considering that the concentration of dormant species in the reaction is approximately $10^2$ to $10^3$ times higher than that of Cu(II)Br$_2$/TPMA (about 5000 ppm for the initiator versus approximately 9 ppm for Cu(II)Br$_2$/TPMA; it is assumed that Cu(II)Br$_2$/TPMA is to be approximately 90% of total copper in the reaction)[96], the reaction mechanism depends on the selection of PCs. For 4Cz-IPN and 4DCDP-IPN, the reaction follows the Mechanism I (Fig. 1a, left), characterized by the reduction of the Cu(II)Br$_2$/L to Cu(II)Br/L, significantly influencing the initiation process. Conversely, for 4DP-IPN, EBPA plays a substantial role in the initiation, suggesting that this reaction proceeds through the Mechanism II (Fig. 1a, center). Moreover, among the three PCs, the rate constants of ET for 4DP-IPN with Cu(II)Br$_2$/TPMA and EBPA are the fastest, followed by 4Cz-IPN, while 4DCDP-IPN has the slowest rate constants. This ordering correlates well within the trends observed in $E_{ox}^*$. Given the generation of the $T_1$ state is much more efficient in 4DP-IPN and 4DCDP-IPN than in 4Cz-IPN, 4DP-IPN exhibits much faster initiation than the other two PCs (Supplementary Fig. 31). However, the slightly better performances of 4Cz-IPN and 4DCDP-IPN at a loading of 50 ppb are likely due to their more positive $E_{ox}^0$ values, which facilitate PC regeneration.

According to the previous literature[53–58], PC regeneration is assumed to primarily occur through the oxidation of TPMA. However, given the redox potentials, not only TPMA but also Cu(II)Br/L with bromide and propagating radical species with bromide can regenerate the PC (see Supplementary Fig. 14 and Supplementary Table 5 for the driving forces ($-\Delta G_{ET}$)). Nonetheless, while these pathways involve three components, PC regeneration through the ligand oxidation involves only two molecules. Furthermore, given the relatively high concentration of the ligand—used in excess compared to the Cu(II)Br/L and the propagating radicals—we agree with other researchers that ligand oxidation is the primary mechanism for the PC regeneration. The quantity of TPMA significantly influences the reaction; when using TPMA in a 1:1 molar ratio with Cu(II)Br$_2$, the reaction did not proceed (Supplementary Table 7). Furthermore, an increase in the quantity of TPMA leads to higher polymerization conversions. This further supports our argument that the ligand-induced regeneration of PC plays critical roles in the polymerization. Direct measurement of the ET rate at which TPMA reduces the PC radical cation (PC$^{•+}$) is challenging due to practical difficulties in an isolation of PC$^{•+}$. Consequently, we instead inferred this by measuring the photoinduced ET rate constant between the PC and TPMA. Figure 3e presents the Stern-Volmer plot between the PCs and TPMA. 4Cz-IPN exhibited a fastest ET rate constant, followed by 4DCDP-IPN, and 4DP-IPN expectedly showed a rate approximately $10^2$ times slower rate constant than the other two PCs; as unanticipated, the behaviors of three PCs slightly mismatched with the trend of $E_{ox}^0$, the phenomenon is not yet fully understood and necessitates further research. Nevertheless, this result unequivocally underscores the significance of the $E_{ox}^0$ of the PC.

Summarizing the mechanistic studies conducted so far, the primary initiation pathway involves the reduction of Cu(II)Br$_2$/L by the PCs to generate Cu(II)Br/L. Consequently, $E_{ox}^*$ of the PCs, while significant, does not play the critical role in the polymerization. Specifically, in the case of PC that readily forms the long-lived $T_1$ state, the rapid initiation can be achieved even with the extremely low loadings of PC, despite the potentially insufficient $E_{ox}^*$. On the other hand, if the regeneration of PC is inefficient, the depletion of PC occurs during the reaction, impeding its progress. Therefore, it seems that the sufficiently high $E_{ox}^0$ is essential for the smooth regeneration of the PC.

## Control in ATRP with photoredox/copper dual catalysis
We investigated the controllability of ATRP using the best-performing PC, 4DCDP-IPN, at 50 ppb under the optimized conditions. Initially, we monitored in-situ monomer conversions of the polymerization using nuclear magnetic resonance (NMR) spectroscopy. The monomer conversion had increased over time following pseudo-first-order kinetics (Fig. 4a). In addition, MW of the polymers increased as a function of the monomer conversion (Fig. 4b), which is a typical observation in CRP. This indicates that the polymerizations were sufficiently well controlled. We here observed deviations between the experimental and theoretical MW of approximately 2000−3000 g $mol^{-1}$, resulting in an initiator efficiency (I*) of less than 100%, which aligns with previously reported instances in the synthesis of PMMA[50,97]. This discrepancy is likely attributed to a high dead-chain fraction[10] caused by irreversible terminations between the propagating radicals in the early stages. Subsequently, we conducted temporal control experiments (Fig. 4c). As anticipated, the monomer conversion was controlled by switching the irradiation on and off, indicating that the photoredox cycle operates the overall polymerization well. A slight increase in the conversion was observed at the onset in light-off periods. This is likely to be due to the remaining Cu(I)Br/L which is generated in the end of the light-on period and continues to engage in the polymerization even during the light-off periods[61,96,98].

We next tested the monomer scope by using commercially available methacrylate-based monomers with various lengths of alkyl chains or alkane-functionalized groups, employing 4DCDP-IPN at loadings of 1 ppm or lower (Fig. 4d and Supplementary Table 18). We achieved high monomer conversions, over approximately 60%, and a narrow MW distribution (Đ < 1.35) across all the methacrylate-based monomers tested. This demonstrates the wide functional group tolerances of the monomers and the versatility of our system for various applications[99]. Subsequently, we synthesized block copolymers containing the methacrylate-based monomers with PMMA macroinitiators ($M_n$ = 7500 and Đ = 1.22), which were prepared by using Cu(II)Br$_2$ at 10 ppm and 4DCDP-IPN at 50 ppb (Fig. 4e). The resulting copolymers showed clear shifts in gel permeation chromatography (GPC) traces towards the higher MW regions compared to the macroinitiators, confirming a high degree of chain-end fidelity. Moreover, we explored the synthesis of block copolymer with MMA and styrene (PMMA-b-PS) (Supplementary Fig. 32). After preparing the PMMA macroinitiator ($M_n$ = 6700 and Đ = 1.22) with 10 ppm of Cu(II)Br$_2$ and 1 ppm of 4DCDP-IPN, we synthesized PMMA-b-PS with $M_n$ = 8800 and Đ = 1.17 using the same amounts of Cu(II)Br$_2$ and PC. These Cu(II)Br$_2$ and PC loadings are significantly lower compared to those in the previously reported systems[50,52], where Cu(II)Br$_2$ and PC were commonly used at 100 ppm and 1000 ppm, respectively, to synthesize PMMA-b-PS.

The advantage of ATRP using photoredox/copper dual catalysis lies in its oxygen tolerance. Indeed, various applications utilizing this method in open-air environments have actively been studied[59]. As reported in the previous studies, this oxygen tolerance is likely achieved through oxygen scavenging by Cu(I)Br/L, which is present at significantly higher concentration than the propagating radicals[59,63–65]. In addition, the photoexcited PC contributes to that by generating reactive oxygen species such as singlet oxygen that subsequently react with amines[59,77,100]. To examine the oxygen tolerances in our system, we conducted the synthesis of PMMA under the optimized conditions using Cu(II)Br$_2$/L at 10 ppm and 4DCDP-IPN at 50 ppb without a degassing process (Table 1, entry 26). As expected, we obtained a well-controlled polymerization with $M_n$ = 19,200 and Đ = 1.22, a degree of control comparable to the degassed results ($M_n$ = 16,700 and Đ = 1.20) (Table 1, entry 25). However, the precise mechanisms behind this oxygen tolerance remain complex and unclear; hence, we are currently investigating them. Furthermore, with these low loadings of Cu(II)Br$_2$/L and PC, the reaction did not proceed under completely the open-air conditions, requiring higher amounts of Cu(II)Br$_2$/L and PC for efficient progress.

## Scale-up experiments

High-capacity, large-scale polymerization is crucial for cost-effective mass production. Recently, attempts have been made to conduct the large-scale ATRP with photoredox/copper dual catalysis using a heterogeneous PC, specifically PPh$_3$-CHCP, and sunlight[62]. However, these attempts were limited by the need for a large amount of Cu(II)Br$_2$ (200 ppm) and PC (0.5 mg mL$^{-1}$), as well as the requirements for preparation under an inert atmosphere. To address these issues, we envisioned our system overcoming these limitations by leveraging its excellent oxygen tolerance and the extremely low levels of PC loading, which allow for deeper light penetration[39]. We conducted a large-scale polymerization of PMMA under the ambient conditions without degassing process (Fig. 4f). Employing Cu(II)Br$_2$ at 10 ppm and 4DCDP-IPN at 50 ppb, we successfully achieved well-controlled polymerization of PMMA with 65% monomer conversion over 24 h and a narrow MW distribution (Đ = 1.23), resulting in the production of colorless PMMA (Fig. 4f, right). Subsequently, we investigated the kinetics of polymerization at a higher PC loading of 1 ppm under the same conditions, observing a notably faster rate of polymerization with 74% monomer conversion achieved within 4 h (Supplementary Fig. 33). However, the MW distribution (Đ = 1.36) exhibited slight broadening. Finally, to confirm that its chain-end fidelity remained intact, we carried out the chain extension experiments (Fig. 4f). Using the macroinitiators prepared by the large-scale polymerization with 4DCDP-IPN at 50 ppb, we successfully prepared PMMA-*b*-PMMA with Đ = 1.28, exhibiting a clear shift to higher MW region in GPC traces. These results indicate that even in the presence of oxygen, our system is highly robust with a comparably low level of Cu(II)Br$_2$ and PC loadings. We hope this scalability contribute to expanding applications in areas such as bio-applications[101–103], membranes[104,105], coatings[106–108], adhesives[109–111], and etc.[112,113].

## Discussion

In summary, we have developed the highly efficient ATRP with dual photoredox/copper catalysis, complemented by thorough the mechanistic and structure-property-performance relationship studies. Based on the careful mechanism studies previously proposed by others, we noticed that this reaction (i) initiates through the reduction of Cu(II)Br$_2$/L by the excited state PC, (ii) is controlled through the ATRP equilibrium, and (iii) its catalytic cycle closes with the regeneration from PC$^{•+}$ into PC by the oxidation of ligand. Through the screening of nine PCs with varying redox potentials and T$_1$ state generation abilities, we established the criteria for the design of highly efficient PCs: (i) a highly positive $E_{ox}^0$ capable of rapid regeneration of PC from the ligand, and (ii) the long-lived T$_1$ state of PC for rapid initiation. Based on these findings, we identified the exceptionally efficient PC, 4DCDP-IPN, featuring the maximized $E_{ox}^0$ and moderate $E_{ox}^*$. Utilizing 4DCDP-IPN at a low loading of 50 ppb, we conducted the controlled polymerization, resulting in the narrow MW distributions, high fidelity of chain-ends, robust functional group tolerances, and oxygen tolerances. Since low PC loading allows for the deeper light penetration, we demonstrated the large-scale polymerizations, under the ambient conditions without a degassing process. We hope this work has positive impacts not only on the CRP but also in various areas such metallaphotoredox-mediated organic syntheses[114–116].

## Methods

### General experimental procedures for polymerization

A typical experimental procedure, for instance, with [MMA]$_0$:[EBPA]$_0$:[PC]$_0$:[Cu(II)Br$_2$]$_0$:[TPMA]$_0$ = [200]:[1]:[0.0002]:[0.002]:[0.009] was carried out as follows. A 20 mL vial (glass, Sungho SIGMA) equipped with a stirring bar was charged with MMA (1.0 mL, 9.29 mmol), EBPA (8.39 μL, 0.046 mmol), PC (0.0093 μmol), Cu(II)Br$_2$ (0.093 μmol), TPMA (0.42 μmol), and anhydrous DMF (1 mL) as a solvent. Prepared stock solutions of the PCs, Cu(II)Br$_2$/TPMA, and the

initiator were used for the higher reproducibility results. After then, the vial was capped with a rubber septum and sealed with parafilm and bubbled with 99.999% N$_2$ for 30 min. Subsequently, polymerizations were carried out under 455 nm irradiation (50 mW cm$^{-2}$) for 24 h at r.t. For the GPC measurements, the in-situ aliquots were diluted in tetrahydrofuran (THF) and purified to remove the remaining Cu(II)Br$_2$ passing by neutral alumina to remove the Cu(II)Br$_2$.

### General experimental procedures for block copolymerization

A typical experimental procedure for PMMA macroinitiator, for instance, with [MMA]$_0$:[EBPA]:[4DCDP-IPN]$_0$:[Cu(II)Br$_2$]$_0$:[TPMA]$_0$ = [100]:[1]:[0.000005]:[0.001]:[0.0045] was carried out as follows. A 20 mL vial equipped with a stirring bar was charged with MMA (2.0 mL, 18.59 mmol), EBPA (33.54 μL, 0.19 mmol), PC (0.00093 μmol), Cu(II)Br$_2$ (0.19 μmol), TPMA (0.84 μmol), and anhydrous DMF (2 mL) as the solvent. Prepared stock solutions of the PCs, Cu(II)Br$_2$/TPMA and the initiator were used for the higher reproducibility results. After then, the vial was capped with a rubber septum and sealed with parafilm and bubbled with 99.999% N$_2$ for 30 min. Subsequently, polymerizations were carried out under the 455 nm irradiation (50 mW cm$^{-2}$) for 24 h at r.t. To isolate a macroinitiator, the reaction mixture was precipitated into methanol (100 mL) dropwise. Subsequent stirring for 30 min followed by vacuum filtration resulted in the dried polymer which can be used as the macroinitiator (M$_n$ = 7500, Đ = 1.22). Subsequently, block copolymerizations were carried out under the reaction conditions, for instance, with [M]$_0$:[PMMA-Br]$_0$:[4DCDP-IPN]$_0$:[Cu(II)Br$_2$]$_0$:[TPMA]$_0$ = [200]:[1]:[0.00001]:[0.002]:[0.009] in anhydrous DMF (2 mL, (Monomer + Macroinitiator)/DMF = 1/4 (w/v)). After then, a 20 mL vial equipped with a stirring bar was capped with a rubber septum and sealed with parafilm and bubbled with 99.999% N$_2$ for 30 min. Subsequently, the polymerizations were carried out under the 455 nm irradiation (50 mW cm$^{-2}$) for 24 h at r.t.

### General experimental procedures for large-scaled polymerization

A typical ATRP large-scale polymerization procedure, for instance, with [MMA]$_0$:[EBPA]$_0$:[4DCDP-IPN]$_0$:[Cu(II)Br$_2$]$_0$:[TPMA]$_0$ = [200]:[1]:[0.00001]:[0.002]:[0.009] was carried out as follows. A 250 mL one-neck round flask equipped with a stirring bar was charged with MMA (100 mL, 929 mmol), EBPA (0.84 mL, 4.65 mmol), PC (0.047 μmol), Cu(II)Br$_2$ (9.29 μmol), TPMA (41.83 μmol), and anhydrous DMF (100 mL) as the solvent. Prepared stock solutions of the PCs and Cu(II)Br$_2$/TPMA were used for the higher reproducibility results. After then, the flask was capped with a rubber septum and sealed with parafilm without any degassing process. Subsequently, the polymerization was carried out under 455 nm irradiation (200 mW cm$^{-2}$) for 24 h at r.t.

## Data availability

The authors declare that the data supporting the findings of this study are available within the paper and its Supplementary Information. The coordinates for the computationally determined structures are provided as a Source Data file. All data may be obtained from the corresponding author upon request. Source data are provided with this paper.

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

## Acknowledgements

This work was supported by the National Research Foundation of Korea (NRF) grant funded by the Korean government (MSIT) (No. 2021R1A5A1030054).

## Author contributions

W.J., Y.K., and M.S.K. were responsible for the initial conception of the project. W.J., Y.K., and M.S.K. were involved in the discussion of the photophysics and polymerizations and were responsible for writing and editing the final manuscript. W.J. performed all the polymerizations, measured the electrochemical and photophysical measurements, and conducted the kinetic simulations and DFT calculations supported by Y.K. W.J. wrote the initial draft of the manuscript. All the authors dis-cussed the results and commented on the manuscript. Y.K and M.S.K. supervised the project.

## Competing interests

The authors declare no competing interests.

## Additional information

**Peer review information** *Nature Communications* thanks Saihu Liao, Bernd Strehmel, and the other, anonymous, reviewer(s) for their con-tribution to the peer review of this work. A peer review file is available.

