## [Peer Review File · Nature Communications]

Highly efficient dual photoredox/copper catalyzed atom transfer radical polymerization achieved through mechanism-driven photocatalyst designReviewers' Comments:

Reviewer #1:

Remarks to the Author:

This manuscript describes a very interesting approach to the sensitized photo-ATRP with the goal of reducing the amount of copper catalyst to a very low level and pursuing the reaction in the presence of oxygen, resulting in a breathing photo-ATRP system. The selection PCs are interesting, and I also like the comparison. The authors also nicely worked out the contribution of triplet states as documented by the long, long lifetimes obtained remaining in the microsecond region. This manuscript was carefully written. I have the opinion that experiments were carefully pursued. Again, a very innovative idea is introduced in this journal. I also like the summary of PCs in the SI that were previously introduced.

However, I have some remarks on which authors should work to improve the quality and make it more friendly for the reader.

- 1) The structure of Eosin Y is wrong. Aldrich has that in the catalog on their websites. It has messed up many papers in A class journal, but again, it is wrong. The structure shows the lactone that exhibits no color. The ring-opened form is colored, as shown by the data. This must be corrected.
- 2) The reaction schemes are hard to read, and I have some doubts about them. According to these schemes, the ligand would oxidize. What will happen with the cation radical? More like is to me, that the cation radical of the sensitizer reacts with bromide and the polymer radical to the PC and CuBr₂ to close the catalytic cycle. Authors should think about their mechanisms.
- 3) The role of oxygen must be better explained. If triplets are formed, there must be singlet oxygen formed. Do they have some evidence for it? It emits at 1280 nm. Singlet oxygen oxidizes well with other substrates, resulting in the consumption of inhibiting triplet oxygen. This needs deeper discussion.
- 4) The discussion is too short. It should be definitively extended.
- 5) The argumentation of the pattern in the ON/OFF cycle is too weak. It also seems that there is some Cu(I) left, which causes polymerization in the dark. This part needs deeper discussion.

Reviewer #2:

Remarks to the Author:

In this manuscript, the authors have developed an efficient ATRP using dual PC/copper catalysis, complemented by mechanistic and structure-property-performance relationship studies. Through the screening of IPN derivatives, including previously used PCs and newly introduced cyanoarene-based PCs, they established the criteria for the selection of PCs. By using the cyanoarene-based 4DCDP-IPN, poly(methyl methacrylate) was successfully synthesized with high conversion and good control, in the presence of as low as 10 ppm of copper complexes combining 50 ppb of photoredox catalysts. This is a nice study and could become suitable for publication after addressing the issues described below:

- 1) There are several known studies using the combination of PC with Cu catalysts at low ppm catalyst loadings, while the most desired part is to decrease the loading of copper catalysts for the ideal goal of metal-free. Therefore, the metal-free ATRP with low/sub ppm catalyst loadings needs to be discussed and mentioned in the introduction to show the state-of-art. E.g. *Macromol. Rapid Commun.* 2017, 38, 1600461 (this study first used IPN-type organic PC for ATRP, and the 4Cz-IPN was also effective in the current study at 50 ppb loadings, entry 22, Table 1); *Nat. Catal.* 2018, 1, 794–804; *Nat. Commun.* 2021, 12, 429, and the latter study showed the successful polymerization with 50 ppb

of organic PC in the absence of copper catalysts. These papers should be mentioned and cited appropriately in the main text.

2) Figure 4, the temporal control experiment with a longer light-off period is necessary to demonstrate the effectiveness of light regulation. For a nice example, please see Figure 3D in a recent JACS paper (J. Am. Chem. Soc. 2021, 143, 25, 963) from the Matyjaszewski group, which showed a good temporal control in the polymerization of MA.

3) The catalytic system showed high efficiency in the polymerization of MMA; can the catalysts promote the polymerization of MA or St at this low catalyst loading?

4) How about the performance of the catalysts in MeCN or DCM? Some monomers are often not compatible with DMF. The results should be included into the main text or supplementary materials, which could be helpful for the readers to choose suitable methods.

Reviewer #3:

Remarks to the Author:

This is a great study by Kwon and coworkers on the development of dual photoredox/copper catalysis in ATRP. The authors took a comprehensive approach in discussing the fundamentals of dual photoredox/copper catalysis and the relevance of the excited state properties of the photocatalyst enabling selective reduction of the copper complexes as opposed to the activation of the alkyl halide initiator. An important aspect of this study is also the discussion on the importance of photoredox catalyst regeneration that requires higher oxidation potentials than that of the amine electron donor (in this case TPMA ligand used in excess). The authors propose that cyanoarenes are an efficient class of photoredox catalysts in copper-catalyzed ATRP. The cyanoarene with dicyanodiphenylamine groups possesses a long excited-state lifetime with reduction potentials suitable for selective reduction of the copper catalyst while also forming a highly oxidizing photocatalyst radical cation species (PC•+) that can efficiently oxidize the amine electron donor to complete the catalytic cycle and regenerate the photocatalyst. The authors show that this photocatalyst enables copper-catalyzed ATRP at concentrations as low as 50 ppb of the photocatalyst under blue light. I think this paper provides a thorough and systematic insight into developing photoredox catalysts for polymerization reactions and beyond, and therefore may be suitable for publication in Nature Communications.

Some minor comments:

- I think in ATRP equilibrium maintaining a low concentration of radicals may not necessarily (or exclusively) lead to controlled polymerization. It is important to establish an efficient and rapid exchange of polymer chains between active and dormant states (via activation and deactivation processes involving halogen atom transfer, respectively) for a controlled polymerization by ATRP catalysis. The reversible deactivation of propagating radicals by L/CuII-Br species to form Br-capped chains is an important aspect of control in ATRP, rather than just keeping the radical concentration low.

- Line 47: Oxygen tolerance in organo-catalyzed ATRP performed in air was investigated previously in ACS Macro Lett. 2018, 7, 8, 1016–1021.

- Line 254: As the authors provide evidence of temporal control only, I would suggest changing 'spatiotemporal control' to 'temporal control'.

- Temporal control: It would also be interesting to see/compare temporal control over a long off time.

- Line 266: '(PMA)' should be changed to '(BMA)'.

- Line 265-266: I am not sure how dispersity of copolymers can be directly correlated with the propagation rate constants of corresponding monomers.

- Please confirm the structure of the copolymers in Figure 4e. The scheme depicts the second monomer in copolymerization as butyl acrylate and benzyl acrylate whereas the text indicates butyl methacrylate (BMA) and benzyl methacrylate (BzMA) (i.e., a methyl group is missing on BMA and BzMA).

- Finally, in Abstract or throughout the manuscript, the authors may provide specific information about the general structure and functionality of the photocatalyst than just stating '4DCDP-IPN', which may not be immediately clear for the readers in understanding its structural properties.

Reviewers' comments to author:

We are grateful to the reviewers for their encouraging comments and constructive suggestions, which greatly helped us to avoid misunderstandings, and to clarify important points of our work. We have carefully revised our manuscript in line with all the reviewers' comments. The point-by-point response is given below. Our responses to reviewers' comments are highlighted in *blue color* with a highlight in the revised manuscript and SI.

Reviewer #1 (Remarks to the Author):

This manuscript describes a very interesting approach to the sensitized photo-ATRP with the goal of reducing the amount of copper catalyst to a very low level and pursuing the reaction in the presence of oxygen, resulting in a breathing photo-ATRP system. The selection PCs are interesting, and I also like the comparison. The authors also nicely worked out the contribution of triplet states as documented by the long, long lifetimes obtained remaining in the microsecond region. This manuscript was carefully written. I have the opinion that experiments were carefully pursued. Again, a very innovative idea is introduced in this journal. I also like the summary of PCs in the SI that were previously introduced.

However, I have some remarks on which authors should work to improve the quality and make it more friendly for the reader.

Response: We appreciate the positive comments from Reviewer #1. Our responses to Reviewer #1's comments are detailed below and reflected in the revised manuscript, where changes are indicated by text in blue with a *yellow* highlight.

1) The structure of Eosin Y is wrong. Aldrich has that in the catalog on their websites. It has messed up many papers in A class journal, but again, it is wrong. The structure shows the lactone that exhibits no color. The ring-opened form is colored, as shown by the data. This must be corrected.

Response: Following the referee's comments, we have corrected the structure of Eosin Y and accurately reflected these changes in *Fig. 2* and *Supplementary Fig.4* of the revised manuscript and SI.

2) The reaction schemes are hard to read, and I have some doubts about them. According to these schemes, the ligand would oxidize. What will happen with the cation radical? More like is to me, that the cation radical of the sensitizer reacts with bromide and the polymer radical to the PC and CuBr₂ to close the catalytic cycle. Authors should think about their mechanisms.

Response: We sincerely appreciate the reviewer's critical comments regarding photocatalyst (PC) regeneration. As noted, the PC regeneration process can occur through three mechanisms: i) $PC^{*+} + L \rightarrow PC + L^{*+}$, ii) $PC^{*+} + R_n^{\bullet} + Br^{-} \rightarrow PC + P-Br$, and iii) $PC^{*+} + Br^{-} + Cu(I)Br/L \rightarrow PC + Cu(II)Br_2/L$. We believe that PC regeneration via an oxidation of the ligand will be the most dominant for following reasons. i) Generally, dual photoredox/copper ATRP systems use the

ligand in a huge excess compared to the PC or Cu(II)Br₂. Under actual conditions, the ligand is used in an amount approximately 1000 times that of the PC and about 5 times that of Cu(II)Br₂. ii) Under the reaction conditions, the actual concentrations of propagating radical species (R_n[•]) and Br⁻ is low. iii) The reaction involving the ligand is a bimolecular process, whereas the other two are termolecular. Indeed, it is crucial to use the excess amount of ligand. As shown in **Supplementary Table 7**, the reaction is highly sensitive to the ligand amount. When the ligand is used at a 1:1 molar ratio with Cu(II)Br₂, the reaction does not proceed. This observation aligns with the findings reported by other groups (*J. Am. Chem. Soc.* **143**, 9630–9638 (2021); *Nat. Commun.* **14**, 2891 (2023)). To clarify these points, we have detailed the mechanism of the dual photoredox/copper ATRP in **Supplementary Fig. 14** of the revised supplementary information (see **Fig. R1**). Additionally, the driving force related to PC regeneration has been added to **Supplementary Table 5**.

The following statement was also added to the main text of the revised manuscript:

“Cu(I)Br/L and propagating radical species, along with bromide, can also promote PC regeneration through termolecular processes. However, considering that PC regeneration via ligand oxidation involves a bimolecular reaction, and noting the ligand’s significantly higher concentrations—due to its excess use—relative to Cu(I)Br/L and propagating radical species, it is broadly recognized that ligand oxidation serves as the dominant mechanism for PC regeneration; for a more detailed discussion of the mechanism, please refer to the supplementary information (Supplementary Fig. 14).” and “However, given the redox potentials, not only TPMA but also Cu(I)Br/L with bromide and propagating radical species with bromide can regenerate the PC (see Supplementary Fig. 14 and Supplementary Table 5 for the driving forces (−ΔG_{ET})). Nonetheless, while these pathways involve three components, PC regeneration through the ligand oxidation involves only two molecules. Furthermore, given the relatively high concentrations of the ligand—used in excess compared to Cu(I)Br/L and the propagating radicals—we agree with other researchers that ligand oxidation is the primary mechanism for PC regeneration. In fact, the quantity of TPMA significantly influences the reaction; when using TPMA in a 1:1 molar ratio with Cu(II)Br₂, the reaction did not proceed (Supplementary Table 7). Furthermore, an increase in the quantity of TPMA leads to higher polymerization conversion. This further supports our argument that ligand-induced regeneration of PC plays critical roles in the polymerization.”

Supplementary Table 7 Results of ATRP with photoredox/copper dual catalysis depending on the amount of ligand.

Entry	PC	PC loading (ppm)	Cu(II)Br ₂ (ppm)	Ligand (ppm)	α (%)	M _{n,theo} (g mol ⁻¹) ^a	M _{n,exp} (g mol ⁻¹) ^b	I* ^c	Đ ^d
1				10		No polymerization			
2				20	53	10,900	11,600	0.94	1.22
3	4DCDP-IPN	0.05	10	45	57	11,700	13,100	0.89	1.21
4				100	64	13,100	14,900	0.88	1.20
5				200	69	14,000	15,200	0.92	1.20

Polymerizations were carried out under the irradiation of 455 nm (50 mW cm⁻²) for 24 h at r.t. Conversion was determined by ¹H NMR. ^aM_{n,theo} = [MMA]₀/[EBPA]₀ × conversion × M_{n,MMA} + M_{n,EBPA}. ^bDetermined by GPC using PMMA standards. ^cI* = M_{n,theo}/M_{n,exp}. ^dĐ = M_{w,exp}/M_{n,exp}. Reaction condition: [MMA]₀: [EBPA]₀: [4DCDP-IPN]₀: [Cu(II)Br₂]₀: [TPMA]₀ = [200]: [1]: [0.00001]: [0.002]: [x], MMA/DMF = 1/1 (v/v), x = 4 × 10⁻², 2 × 10⁻², 9 × 10⁻³, 4 × 10⁻³ or 2 × 10⁻³ (200 ppm, 100 ppm, 45 ppm, 20 ppm and 10 ppm relative to the monomer, respectively).

Supplementary Table 5 Experimental evaluation of driving forces ($-\Delta G_{ET}$) for ET process.

$-\Delta G_{ET}$ (eV)	4DMDP-IPN		4DP-IPN		4tCz-IPN		4Cz-IPN		4DCDP-IPN	
	$^1PC^*$	$^3PC^*$	$^1PC^*$	$^3PC^*$	$^1PC^*$	$^3PC^*$	$^1PC^*$	$^3PC^*$	$^1PC^*$	$^3PC^*$
$PC^* + Cu(II)Br_2/TPMA \rightarrow PC^{++} + Cu(I)Br/TPMA + Br^-$	1.37	1.23	1.19	1.03	1.06	1.06	0.88	0.84	0.67	0.50
$PC^* + R-Br \rightarrow PC^{++} + R^* + Br^-$ (R-Br = EBPA)	0.28	0.14	0.10	-0.06	-0.03	-0.03	-0.21	-0.25	-0.42	-0.59
$PC^* + TPMA \rightarrow PC^{-\cdot} + TPMA^{+\cdot}$	-0.31	-0.45	-0.20	-0.36	0.31	0.31	0.45	0.41	0.46	0.29
$PC^{++} + TPMA \rightarrow PC + TPMA^{+\cdot}$	-0.23		0.02		0.31		0.53		0.67	
$PC^{++} + Cu(I)Br/TPMA + Br^- \rightarrow PC + Cu(II)Br_2/TPMA$	1.01		1.26		1.55		1.77		1.91	
$PC^{++} + R^* + Br^- \rightarrow PC + R-Br$	2.10		2.35		2.64		2.86		3.00	

The driving forces for each ET process were calculated using the following equations: $-\Delta G_{ET} = E_{red}^*(PC^*/PC^*) - E_{ox}^0(Sub^{++}/Sub)$ and $-\Delta G_{ET} = E_{red}^0(Sub/Sub^{\cdot-}) - E_{ox}^*(PC^{++}/PC^*)$ for photoinduced ET process; $-\Delta G_{ET} = E_{red}^0(Sub/Sub^{\cdot-}) - E_{ox}^0(PC^{++}/PC)$ for the ET process in PC regeneration (*Chem. Rev.* **116**, 10075–10166 (2016)). Ground- and excited redox potentials of all PCs were referred from Supplementary Table 3. The ground state redox potentials of ATRP components were estimated as $E_{ox}^0 = 0.99$ V for TPMA, $E_{red}^0 = -0.25$ V for $Cu(II)Br_2/TPMA$, and $E_{red}^0 = -1.34$ V for EBPA.

Fig. R1 Proposed reaction mechanisms including reaction pathways of (a) the ATRP with dual photoredox/copper catalysis, (b) a ligand regeneration and (c) a side reaction from ligand radical cation ($L^{+\cdot}$).

3) The role of oxygen must be better explained. If triplets are formed, there must be singlet oxygen formed. Do they have some evidence for it? It emits at 1280 nm. Singlet oxygen oxidizes well with other substrates, resulting in the consumption of inhibiting triplet oxygen. This needs deeper discussion.

Response: We thank the reviewer for the valuable comment. Indeed, we have recently confirmed that cyanoarene-based PCs used in this study (i.e., 4DP-IPN, 4Cz-IPN, and 4DCDP-IPN) can generate singlet oxygen ($^1\text{O}_2$) under light-irradiation conditions in the presence of oxygen, using a chemical $^1\text{O}_2$ detector, specifically 9,10-dimethylanthracene (*Adv. Mater.* **35**, 2204776 (2023); see **Fig. R2**). The figure illustrates that under 455 nm irradiation, all three PCs facilitate the formation of 9,10-endoperoxide, thereby confirming $^1\text{O}_2$ generation through energy transfer from the triplet excited (T_1) states. The generated $^1\text{O}_2$, when reacted with an excess amount of amine, can contribute to increased oxygen tolerance (*Green Chem.* **13**, 3341–3344 (2011)). Additionally, it has been reported that Cu(I)Br/L, which can be continuously regenerated by PC, consumes O_2 to confer high oxygen tolerance (*Chem. Sci.* **13**, 11540–11550 (2022)). However, the detailed mechanism behind the oxygen tolerance remains still unclear. Our group is conducting further investigations to understand the mechanism of oxygen tolerance more clearly.

We have incorporated these contents into the revised manuscript as follows:

“The mechanism for remarkable oxygen tolerance is believed to rely on the oxygen scavenging ability of Cu(I)Br/L and/or the lowest triplet excited (T_1) state of PC.^{55,58}” and “As reported in the previous studies, this oxygen tolerance is likely achieved through oxygen scavenging by Cu(I)Br/L, which is present at significantly higher concentration than the radicals.^{58,61–63} Additionally, the photoexcited PC contributes to that by generating reactive oxygen species such as singlet oxygen that subsequently react with amines.^{58,75,98} To examine the oxygen tolerances in our system, we conducted the synthesis of PMMA under the optimized conditions using Cu(II)Br₂/L at 10 ppm and 4DCDP-IPN at 50 ppb without a degassing process (Table 1, entry 26). As expected, we obtained a well-controlled polymerization with $M_n = 19,200$ and $\bar{D} = 1.22$, a degree of control comparable to the degassed results ($M_n = 16,700$ and $\bar{D} = 1.20$) (Table 1, entry 25). However, the precise mechanisms behind this oxygen tolerance remain complex and unclear; hence, we are currently investigating them.”

Fig. R2 Detection of singlet oxygen ($^1\text{O}_2$) generated by PCs was conducted under the irradiation of 455 nm (100 mW cm⁻²) using ethyl acetate as the solvent in an undegassed condition. The experiment involved (a) 4DP-IPN, (b) 4Cz-IPN, and (c) 4DCDP-IPN, each at a concentration of 1×10^{-5} M, along with 9,10-dimethylanthracene at a concentration of 1×10^{-4} M. Contents of **Fig. R2** are reproduced from cited publications (*Adv. Mater.* **35**, 2204776 (2023)).

4) The discussion is too short. It should be definitively extended.

Response: Most of the discussion has been addressed in the Results section through systematic experimental designs and detailed explanations, following the format of 'Nature Communications'. In any case, in response to the reviewer's comments, we have greatly strengthened the discussion section, which corresponds to the Conclusion in the 'Nature Communications' format, as follows:

“In summary, we have developed a highly efficient ATRP with dual photoredox/copper catalysis, complemented by thorough mechanistic and structure-property-performance relationship studies. Based on the careful mechanism studies previously proposed by others, we noticed that this reaction i) initiates through the reduction of Cu(II)Br₂/L by the excited state PC, ii) is controlled through ATRP equilibrium, and iii) its catalytic cycle closes with the regeneration from PC⁺ into PC by the oxidation of ligand. Through the screening of nine PCs with varying redox potentials and T₁ state generation abilities, we established the criteria for the design of a highly efficient PC: i) a highly positive E_{ox}⁰ capable of rapid regeneration of PC from ligand, and ii) the long-lived T₁ state of PC for rapid initiation. Based on these findings, we identified an exceptionally efficient PC, 4DCDP-IPN, featuring the maximized E_{ox}⁰ and moderate E_{ox}^{}. Utilizing 4DCDP-IPN at a low loading of 50 ppb, we conducted the controlled polymerization, resulting in narrow MW distributions, high fidelity of chain-ends, robust functional group tolerances and oxygen tolerances. Since low PC loading allows for the deeper light penetration, we demonstrated large-scale polymerizations, under ambient conditions without a degassing process. We hope this work has positive impacts not only on CRP but also in various areas such metallaphotoredox-mediated organic syntheses.^{112–114}”*

5) The argumentation of the pattern in the ON/OFF cycle is too weak. It also seems that there is some Cu(I) left, which causes polymerization in the dark. This part needs deeper discussion.

Response: Following the reviewer's comment, to clearly observe the on/off cycle, we conducted experiments with longer intervals (**Fig. R3**). From the experiments, we can confirm that temporal control is effectively achieved. However, we noted slight increases in conversion at the beginning of light-off phase. These phenomena were attributed to the presence of Cu(I)Br/L even in the dark—9.8% in the ATRP equilibrium with TPMA (*Macromolecules* **53**, 5280–5288 (2020); *J. Polym. Sci. Part A Polym. Chem.* **57**, 268–273 (2019)). Such increases in conversion have indeed seen in the previous literature (e.g., *J. Am. Chem. Soc.* **143**, 9630–9638 (2021); *Macromolecules* **56**, 4181–4189 (2023)). Based on this, we have properly revised **Fig. 4c** and added the discussion for the presence of Cu(I) in main manuscript as follows:

“A slight increase in conversion was observed at the onset in light-off periods. This is likely to be due to the remaining Cu(I)Br/L which is generated in the end of light-on period and continues to engage in the polymerization even during the light-off periods.^{59,94,96}”

Fig. R3 (a and b) Temporal control of synthesis of PMMA by switching irradiation on and off. Polymerizations were carried out under the irradiation of 455 nm (50 mW cm^{-2}) at r.t. following the general procedure with condition, $[\text{MMA}]_0:[\text{EBPA}]_0:[\text{4DCDP-IPN}]_0:[\text{Cu(II)Br}_2]_0:[\text{TPMA}]_0 = [200]:[1]:[0.00001]:[0.002]:[0.009]$, $\text{MMA/DMF} = 1/1$ (v/v).

Reviewer #2 (Remarks to the Author):

In this manuscript, the authors have developed an efficient ATRP using dual PC/copper catalysis, complemented by mechanistic and structure-property-performance relationship studies. Through the screening of IPN derivatives, including previously used PCs and newly introduced cyanoarene-based PCs, they established the criteria for the selection of PCs. By using the cyanoarene-based 4DCDP-IPN, poly(methyl methacrylate) was successfully synthesized with high conversion and good control, in the presence of as low as 10 ppm of copper complexes combining 50 ppb of photoredox catalysts. This is a nice study and could become suitable for publication after addressing the issues described below:

Response: We are grateful for the highly encouraging comments from Reviewer #2. Our responses to reviewer #2' comments are given below highlighted as **green**.

1) There are several known studies using the combination of PC with Cu catalysts at low ppm catalyst loadings, while the most desired part is to decrease the loading of copper catalysts for the ideal goal of metal-free. Therefore, the metal-free ATRP with low/sub ppm catalyst loadings needs to be discussed and mentioned in the introduction to show the state-of-art. E.g. *Macromol. Rapid Commun.* 2017, 38, 1600461 (this study first used IPN-type organic PC for ATRP, and the 4Cz-IPN was also effective in the current study at 50 ppb loadings, entry 22, Table 1); *Nat. Catal.* 2018, 1, 794–804; *Nat. Commun.* 2021, 12, 429, and the latter study showed the successful polymerization with 50 ppb of organic PC in the absence of copper catalysts. These papers should be mentioned and cited appropriately in the main text.

Response: We appreciate the reviewer's comment. Following the reviewer's comment, we have added the relevant information to the introduction and appropriately cited it as follows:

“By carefully designing the PCs,^{38–40} the range of light capable of initiating the reaction can be expanded to include visible light, and the PC loadings can also be reduced to the level of ppm.³⁸ For example, the groups led by Zhu and Miyake reduced PC loadings to approximately 10 ppm for the synthesis of poly(methyl methacrylate) (PMMA).^{41,42} Later, the groups of Kwon and Liao further reduced PC loadings to sub-ppm levels.^{39,43}”

2) Figure 4, the temporal control experiment with a longer light-off period is necessary to demonstrate the effectiveness of light regulation. For a nice example, please see Figure 3D in a recent JACS paper (*J. Am. Chem. Soc.* 2021, 143, 25, 963) from the Matyjaszewski group, which showed a good temporal control in the polymerization of MA.

Response: Following the reviewer's comment, to clearly observe the on/off cycle, we conducted experiments with longer intervals (**Fig. R4**). From the experiments, we can confirm that temporal control is effectively achieved. However, we noted slight increases in conversion at the beginning of light-off phase. These phenomena were attributed to the presence of Cu(I)Br/L even in the dark—9.8% in the ATRP equilibrium with TPMA (*Macromolecules* **53**, 5280–5288 (2020); *J. Polym. Sci. Part A Polym. Chem.* **57**, 268–273 (2019)). Such increases in conversion have indeed seen in the previous literature (e.g., *J. Am. Chem. Soc.* **143**, 9630–9638 (2021); *Macromolecules* **56**, 4181–4189 (2023)). Based on this, we have properly revised **Fig. 4c** and added the discussion for the presence of Cu(I) in main manuscript as follows:

“A slight increase in conversion was observed at the onset in light-off periods. This is likely to be due to the remaining Cu(I)Br₂/L which is generated in the end of light-on period and continues to engage in the polymerization even during the light-off periods.”^{59,94,98}”

Fig. R4 (a and b) Temporal control of synthesis of PMMA by switching irradiation on and off. Polymerizations were carried out under the irradiation of 455 nm (50 mW cm^{-2}) at r.t. following the general procedure with condition, $[MMA]_0:[EBPA]_0:[4DCDP-IPN]_0:[Cu(II)Br_2]_0:[TPMA]_0 = [200]:[1]:[0.00001]:[0.002]:[0.009]$, $MMA/DMF = 1/1$ (v/v).

3) The catalytic system showed high efficiency in the polymerization of MMA; can the catalysts promote the polymerization of MA or St at this low catalyst loading?

Response: Following the reviewer’s comments, to check the feasibility into synthesis of polystyrene (PSt) and poly(methyl acrylate) (PMA), we conducted the polymerization of St (**Table R1**) and MA (**Table R2**) under same reaction condition with PMMA synthesis. In both cases, the results were inferior to those of PMMA synthesis. Specifically, the synthesis of PSt exhibit low conversion compared to the PMMA synthesis. This would be because the propagating rate constant (k_p) of St is very slow ($k_p = 108 \text{ M}^{-1} \text{ s}^{-1}$) (*Chemical Reviews* **122**, 1830–1874 (2022)), thus an increase in the reaction temperature and ligand optimization would be required to achieve a higher monomer conversion with decent controllability (*Nat. Catal.* **1**, 794-804 (2018); *Nat. Commun.* **14**, 2891 (2023)). In the case of PMA synthesis, we observed decreased monomer conversion and inefficient initiator efficiency at lower PC concentrations (i.e., 0.1 and 0.05 ppm). Because the k_p of MA is very faster ($k_p = 14,800 \text{ M}^{-1} \text{ s}^{-1}$) than that of MMA ($k_p = 369 \text{ M}^{-1} \text{ s}^{-1}$) (*Chemical Reviews* **122**, 1830–1874 (2022)), the usage of TPMA as the ligand is mismatched for PMA. Therefore, it is required to engage other ligand (e.g., Me₆TREN, (Tris[2-(dimethylamino)ethyl]amine)) with a fast k_{act} and/or a higher concentration of Cu(II)Br₂/L in the ATRP equilibrium (*J. Am. Chem. Soc.* **144**, 15413–155430 (2022)).

The variations in reactivity of different monomers need the individual optimized conditions such the kinds of ligands for the appropriate K_{atrp} . Additionally, it is expected that the optimized PC would be differed according to the usage of different ligand. Further investigations to expand monomer scope, especially for the synthesis of PSt and PMA, are currently underway. We have added the polymerization results to the SI (**Supplementary Table 19** and **20**).

Table R1. Results of the polystyrene synthesis.
Entry	PC	PC loading (ppm)	Cu(II)Br ₂ (ppm)	Time (h)	α (%)	$M_{n,theo}$ (g mol ⁻¹) ^b	$M_{n,exp}$ (g mol ⁻¹) ^c	I^* ^d	\mathcal{D} ^e
1	4DCDP-IPN	10	100	24	18	4,100	4,800	0.85	1.12
2 ^a	4DCDP-IPN	10	100	24	32	6,900	8,200	0.84	1.20
3	4DCDP-IPN	10	10	24	45	9,300	12,500	0.74	1.62
4	4DCDP-IPN	5	10	24	40	8,200	12,600	0.65	1.81
5	4DCDP-IPN	1	10	24	17	3,800	4,200	0.90	1.21
6				48	31	6,600	6,400	1.03	1.23
7 ^a	4DCDP-IPN	1	10	24	32	6,900	6,600	1.05	1.36

Polymerizations were carried out under the irradiation of 455 nm (50 mW cm⁻²) for 24 h at r.t. following the general procedure with condition, [St]₀:[EBPA]₀:[PC]₀:[Cu(II)Br₂]₀:[TPMA]₀ = [200]:[1]:[x]:[y]:[4.5y], MMA/DMF = 1/1 (v/v) where x = 2 × 10⁻³, 1 × 10⁻³ or 2 × 10⁻⁴ (for 10 ppm, 5 ppm, and 1 ppm relative to the monomer, respectively), y = 2 × 10⁻² or 2 × 10⁻³ (for 100 ppm and 10 ppm relative to the monomer, respectively). Monomer conversion (α) was determined by integrals of ¹H NMR peaks. ^aReaction was performed at temperature of 35 °C. ^b $M_{n,theo}$ was determined by following relationship, $M_{n,theo} = [St]_0/[EBPA]_0 \times \text{conversion} \times M_{n,St} + M_{n,EBPA}$. ^c $M_{n,exp}$ was measured by GPC equipped with refractive index detector calibrated with PMMA standards. ^dInitiator efficiency (I^*) was determined by $I^* = M_{n,theo}/M_{n,exp}$. ^eDispersity (\mathcal{D}) was determined by $\mathcal{D} = M_{w,exp}/M_{n,exp}$.

Table R2. Results of the poly(methyl acrylate) synthesis.
Entry	PC	PC loading (ppm)	Cu(II)Br ₂ (ppm)	α (%)	$M_{n,theo}$ (g mol ⁻¹) ^a	$M_{n,exp}$ (g mol ⁻¹) ^b	I^* ^c	\mathcal{D} ^d
1	4DCDP-IPN	1	10	83	14,600	14,200	1.03	1.47
2	4DCDP-IPN	0.1	10	17	3,200	3,900	0.82	1.26
3	4DCDP-IPN	0.05	10	13	2,400	4,900	0.49	1.35

Polymerizations were carried out under the irradiation of 455 nm (50 mW cm⁻²) for 24 h at r.t. following the general procedure with condition, [MA]₀:[EBPA]₀:[PC]₀:[Cu(II)Br₂]₀:[TPMA]₀ = [200]:[1]:[x]:[0.002]:[0.009], MA/DMF = 1/1 (v/v) where x = 2 × 10⁻⁴, 2 × 10⁻⁵ or 1 × 10⁻⁵ (for 1 ppm, 0.1 ppm and 50 ppb relative to the monomer, respectively). Monomer conversion (α) was determined by integrals of ¹H NMR peaks. ^a $M_{n,theo}$ was determined by following relationship, $M_{n,theo} = [MA]_0/[EBPA]_0 \times \text{conversion} \times M_{n,MA} + M_{n,EBPA}$. ^b $M_{n,exp}$ was measured by GPC equipped with refractive index detector calibrated with PMMA standards. ^cInitiator efficiency (I^*) was determined by $I^* = M_{n,theo} / M_{n,exp}$. ^dDispersity (\mathcal{D}) was determined by $\mathcal{D} = M_{w,exp} / M_{n,exp}$.

4) How about the performance of the catalysts in MeCN or DCM? Some monomers are often not compatible with DMF. The results should be included into the main text or supplementary materials, which could be helpful for the readers to choose suitable methods.

Response: Following the reviewer's suggestion, we conducted additional experiments in acetonitrile (ACN) and dichloromethane (DCM). Polymerization in ACN and DCM exhibited comparable performances at low concentrations of 1 ppm (**Table R3**). In ACN, the kinetics were inferior compared to those in DMF (**Table R3**, entries 4–6), likely due to differences in the K_{trp} influenced by the solvent (*Macromolecules* **42**, 6348–6360 (2009); *Macromolecules* **53**, 4678–4684 (2020); *Nat. Commun.* **14**, 2891 (2023)). Among the three solvents (i.e., DMF, ACN, and DCM), DCM showed the lowest monomer conversion (**Table R3**), as solvents with higher polarity tend to exhibit greater K_{trp} (*Macromolecules* **42**, 6348–6360 (2009); *Macromolecules* **53**, 4678–4684 (2020)). Note that in the case of DCM, controlling the reaction was difficult due to the solvent's volatility; for this reason, the reaction was conducted in a glove box. Cyanoarene-based PCs exhibit strong charge-transfer characteristics in their excited states. Consequently, both their excited state lifetimes and redox potentials are sensitive to a solvent polarity, which may be another reason for the observed solvent dependence. We have added the polymerization results to the supplementary information as **Supplementary Table 21**.

Table R3. Results of the polymerizations with various solvents.

Entry	PC	PC loading (ppm)	Cu(II)Br ₂ (ppm)	Solvent	α (%)	$M_{n,theo}$ (g mol ⁻¹) ^b	$M_{n,exp}$ (g mol ⁻¹) ^c	I^* ^d	\mathcal{D} ^e
1		1			89	18,100	20,300	0.89	1.35
2	4DCDP-IPN	0.1	10	DMF	85	17,200	17,900	0.96	1.21
3		0.05			71	14,400	16,700	0.86	1.20
4		1			89	18,100	17,900	1.01	1.24
5	4DCDP-IPN	0.1	10	ACN	51	10,400	10,400	1.00	1.16
6		0.05			36	7,500	7,900	0.95	1.16
7		1			70	14,300	12,300	1.16	1.19
8	4DCDP-IPN	0.1	10	DCM	28	5,800	4,700	1.23	1.20
9		0.05			20	4,300	3,800	1.13	1.19

Polymerizations were carried out under the irradiation of 455 nm (50 mW cm⁻²) for 24 h at r.t. following the general procedure with condition, [MMA]₀:[EBPA]₀:[4DCDP-IPN]₀:[Cu(II)Br₂]₀:[TPMA]₀ = [200]:[1]:[x]:[0.002]:[0.009], MMA/Solvent = 1/1 (v/v) where x = 2 × 10⁻⁴, 2 × 10⁻⁵ or 1 × 10⁻⁵ (for 1 ppm, 0.1 ppm and 50 ppb relative to the monomer, respectively). Monomer conversion (α) was determined by integrals of ¹H NMR peaks. ^aReaction was performed under undegassed conditions with 2 mL scale using into a 4 mL vial. ^b $M_{n,theo}$ was determined by following relationship, $M_{n,theo} = [MMA]_0/[EBPA]_0 \times \text{conversion} \times M_{n,MMA} + M_{n,EBPA}$. ^c $M_{n,exp}$ was measured by GPC equipped with refractive index detector calibrated with PMMA standards. ^dInitiator efficiency (I^*) was determined by $I^* = M_{n,theo}/M_{n,exp}$. ^eDispersity (\mathcal{D}) was determined by $\mathcal{D} = M_{w,exp}/M_{n,exp}$.

Reviewer #3 (Remarks to the Author):

This is a great study by Kwon and coworkers on the development of dual photoredox/copper catalysis in ATRP. The authors took a comprehensive approach in discussing the fundamentals of dual photoredox/copper catalysis and the relevance of the excited state properties of the photocatalyst enabling selective reduction of the copper complexes as opposed to the activation of the alkyl halide initiator. An important aspect of this study is also the discussion on the importance of photoredox catalyst regeneration that requires higher oxidation potentials than that of the amine electron donor (in this case TPMA ligand used in excess). The authors propose that cyanoarenes are an efficient class of photoredox catalysts in copper-catalyzed ATRP. The cyanoarene with dicyanodiphenylamine groups possesses a long excited-state lifetime with reduction potentials suitable for selective reduction of the copper catalyst while also forming a highly oxidizing photocatalyst radical cation species (PC^{•+}) that can efficiently oxidize the amine electron donor to complete the catalytic cycle and regenerate the photocatalyst. The authors show that this photocatalyst enables copper-catalyzed ATRP at concentrations as low as 50 ppb of the photocatalyst under blue light. I think this paper provides a thorough and systematic insight into developing photoredox catalysts for polymerization reactions and beyond, and therefore may be suitable for publication in Nature Communications.

Response: We are thankful for the exceptionally positive comments from Reviewer #3. Our responses to reviewer #3' comments are given below highlighted as cyan.

Some minor comments:

1) I think in ATRP equilibrium maintaining a low concentration of radicals may not necessarily (or exclusively) lead to controlled polymerization. It is important to establish an efficient and rapid exchange of polymer chains between active and dormant states (via activation and deactivation processes involving halogen atom transfer, respectively) for a controlled polymerization by ATRP catalysis. The reversible deactivation of propagating radicals by L/CuII-Br species to form Br-capped chains is an important aspect of control in ATRP, rather than just keeping the radical concentration low.

Response: Thanks for the reviewer's constructive comment. Following the reviewer's suggestion, we have revised the text as follows:

"ATRP regulates the reaction by an equilibrium between active and dormant species mediated by the activator and deactivator forms of the catalyst, based on the persistent radical effect.^{9,10}"

2) Line 47: Oxygen tolerance in organo-catalyzed ATRP performed in air was investigated previously in ACS Macro Lett. 2018, 7, 8, 1016–1021.

Response: We have properly revised the text and cited reference as follows:

"However, the range of applicable monomers, solvents, and MWs remains relatively limited. Moreover, although there are some recent reports on oxygen tolerance,^{42,44} most studies report no such tolerance."

3) Line 254: As the authors provide evidence of temporal control only, I would suggest changing ‘spatiotemporal control’ to ‘temporal control’.

Response: Following the reviewer’s comments, we have properly corrected.

4) Temporal control: It would also be interesting to see/compare temporal control over a long off time.

Response: Following the reviewer’s comment, to clearly observe the on/off cycle, we conducted experiments with longer intervals (**Fig. R5**). From the experiments, we can confirm that temporal control is effectively achieved. However, we noted slight increases in conversion at the beginning of light off phase. These phenomena were attributed to the presence of Cu(I)Br/L even in the dark—9.8% in the ATRP equilibrium with TPMA (*Macromolecules* **53**, 5280–5288 (2020) and *J. Polym. Sci. Part A Polym. Chem.* **57**, 268–273 (2019)). Such increases in conversion have indeed seen in the previous literature (e.g., *J. Am. Chem. Soc.* **143**, 9630–9638 (2021) and *Macromolecules* **56**, 4181–4189 (2023)). Based on this, we have properly revised **Fig. 4c** and added the discussion for the presence of Cu(I) in main manuscript as follows:

“A slight increase in conversion was observed at the onset in light-off periods. This is likely to be due to the remaining Cu(I)Br/L which is generated in the end of light-on period and continues to engage in the polymerization even during the light-off periods.”^{59,94,96}

Fig. R5 (a and b) Temporal control of synthesis of PMMA by switching irradiation on and off. Polymerizations were carried out under the irradiation of 455 nm (50 mW cm⁻²) at r.t. following the general procedure with condition, [MMA]₀:[EBPA]₀:[4DCDP-IPN]₀:[Cu(II)Br₂]₀:[TPMA]₀ = [200]:[1]:[0.00001]:[0.002]:[0.009], MMA/DMF = 1/1 (v/v).

5) Line 266: ‘(PMA)’ should be changed to ‘(BMA)’.

Response: Following the reviewer’s comments, we have corrected the typo.

6) Line 265-266: I am not sure how dispersity of copolymers can be directly correlated with the propagation rate constants of corresponding monomers.

Response: We appreciate the reviewer's careful comments. As the reviewer noted, the propagating rate constants of methacrylates do not significantly deviate; furthermore, the correlation between the propagation rate constants of monomers and the controllability of chain extension remains unclear, according to some publications (e.g., *Science* **352**, 1082–1086 (2016); *Nat. Commun.* **12**, 429 (2021)). Therefore, to avoid any misunderstanding by the reader, we have removed the statement regarding this correlation from 'Control in ATRP with photoredox/copper dual catalysis.'

“The resulting copolymers showed clear shifts in gel permeation chromatography (GPC) traces towards the higher MW regions compared to the macroinitiators, confirming a high degree of chain-end fidelity.”

7) Please confirm the structure of the copolymers in Figure 4e. The scheme depicts the second monomer in copolymerization as butyl acrylate and benzyl acrylate whereas the text indicates butyl methacrylate (BMA) and benzyl methacrylate (BzMA) (i.e., a methyl group is missing on BMA and BzMA).

Response: Following the reviewer's comments, we have re-drawn their chemical structure.

8) Finally, in Abstract or throughout the manuscript, the authors may provide specific information about the general structure and functionality of the photocatalyst than just stating '4DCDP-IPN', which may not be immediately clear for the readers in understanding its structural properties.

Response: We thank the reviewer for the careful comment. To clearly provide the novelty of our work, we have revised the abstract adding the statement on the structural design for 4DCDP-IPN as follows:

“Through studying polymerization mechanisms, we found that efficient polymerizations are driven by PCs whose ground state oxidation potential—responsible for the regeneration of the PC—plays a more important role than their excited state reducing power, responsible for initiation. This was verified by screening PCs with varying redox potentials and triplet excited state generation capabilities.”

Reviewers' Comments:

Reviewer #1:

Remarks to the Author:

The authors followed my suggestions accordingly. From my point of view, this is a nice contribution. Although the structure looks a bit exotic to achieve such a performance, it may even be necessary to synthesize such stuff to achieve such good results.

There is only one point that might need additional attention. The authors also varied the concentration of PC. Would it be possible to compare the absorptivities? This may be a bit difficult to calculate because they typically use Schlenk tubes. They are round and much harder to modulate than rectangular cells. In other words, absorption is important to reveal photochemical reactivity. If these are not clear and consistent, then additional interactions between the photocatalyst and the metal cocatalyst would be necessary. Authors should check this and either pursue the addition themselves. If this is too difficult, it would be ok for me. Many other papers also show missing parts here. Nevertheless, I have the impression that this manuscript and the experiments have been prepared with great care, and I do not want to criticize the authors for something that others could not follow.

I recommend publication.

Reviewer #2:

Remarks to the Author:

The revised manuscript is OK for publication. Congratulations!

Reviewer #3:

Remarks to the Author:

The authors have properly revised the manuscript and addressed the comments by the reviewers. Therefore, the manuscript should be suitable for publication in Nature Communications at this stage.

Reviewer #1 (Remarks to the Author):

The authors followed my suggestions accordingly. From my point of view, this is a nice contribution. Although the structure looks a bit exotic to achieve such a performance, it may even be necessary to synthesize such stuff to achieve such good results.

Response: We sincerely appreciate the very positive response from the reviewers. In this revised version, we have addressed reviewer's remaining comment. We are grateful for the continued efforts supporting our work to be published.

There is only one point that might need additional attention. The authors also varied the concentration of PC. Would it be possible to compare the absorptivities? This may be a bit difficult to calculate because they typically use Schlenk tubes. They are round and much harder to modulate than rectangular cells. In other words, absorption is important to reveal photochemical reactivity. If these are not clear and consistent, then additional interactions between the photocatalyst and the metal cocatalyst would be necessary. Authors should check this and either pursue the addition themselves. If this is too difficult, it would be ok for me. Many other papers also show missing parts here. Nevertheless, I have the impression that this manuscript and the experiments have been prepared with great care, and I do not want to criticize the authors for something that others could not follow.

Response: Thank you for the reviewer's suggestions. Using rectangular cells, we first measured the UV-vis absorption spectra of 4DCDP-IPN solutions at its various concentrations at a fixed Cu(II)Br₂/TPMA loading of 10 ppm (**Fig. R1**). To minimize the effect of impurities in TPMA (Macromol. Rapid Commun. **44**, 2200855 (2023)), we used a 1:1 molar ratio of Cu(II)Br₂ to TPMA, unlike the actual reaction solution. Consequently, clear and consistent linear correlations between PC concentration and its absorbance were observed as illustrated in inset of **Fig. R1**. Additionally, we further confirmed that no interaction between PC and Cu(II)Br₂/TPMA to induce non-linear correlation was observed (inset of **Fig. R1**, dark solid line and grey dash line). These measured absorbances are consistent with the conversion trend in polymerization at same reaction time, however, to reveal the exact relationships between the absorptivity and polymerization within the actual reaction batch, we have difficulties such batch's curvature, and they would be addressed in follow-up studies. We have added the UV-vis absorption spectra to the SI (**Supplementary Fig. 15**).

Fig. R1 UV-vis absorption spectra of PC solutions varying concentrations in the presence of Cu(II)Br₂/TPMA. PC solutions were prepared by diluted 4DCDP-IPN solutions in DMF as 50 ppm (relative to monomer mimicking our

reaction conditions), 25 ppm, 10 ppm, 5 ppm, and 1 ppm correspond to 230 μM , 115 μM , 46 μM , 23 μM , and 4.6 μM respectively. To prepare $\text{Cu(II)Br}_2/\text{TPMA}$ solution, 1:1 molar ratio of Cu(II)Br_2 and TPMA were used. The correlation between absorbance and PC concentration with (dark solid line) or without (grey dash line) $\text{Cu(II)Br}_2/\text{TPMA}$ were also given (inset).